# VipAct: Visual-Perception Enhancement via Specialized VLM Agent Collaboration and Tool-use

## Abstract

While vision-language models (VLMs) have demonstrated remarkable performance across various tasks combining textual and visual information, they continue to struggle with fine-grained visual perception tasks that require detailed pixel-level analysis. Effectively eliciting comprehensive reasoning from VLMs on such intricate visual elements remains an open challenge. In this paper, we present VipAct, an agent framework that enhances VLMs by integrating multi-agent collaboration and vision expert models, enabling more precise visual understanding and comprehensive reasoning. VipAct consists of an orchestrator agent, which manages task requirement analysis, planning, and coordination, along with specialized agents that handle specific tasks such as image captioning and vision expert models that provide high-precision perceptual information. This multi-agent approach allows VLMs to better perform fine-grained visual perception tasks by synergizing planning, reasoning, and tool use. We evaluate VipAct on benchmarks featuring a diverse set of visual perception tasks, with experimental results demonstrating significant performance improvements over state-of-the-art baselines across all tasks. Furthermore, comprehensive ablation studies reveal the critical role of multi-agent collaboration in eliciting more detailed System-2 reasoning and highlight the importance of image input for task planning. Additionally, our error analysis identifies patterns of VLMs' inherent limitations in visual perception, providing insights into potential future improvements. VipAct offers a flexible and extensible framework, paving the way for more advanced visual perception systems across various real-world applications.

## 1 Introduction

Recent advancements in large multimodal models (LMMs), particularly vision-language models (VLMs) (OpenAI, 2024; Bai et al., 2023; Chen et al., 2024b), have demonstrated remarkable capabilities in tasks that integrate textual and visual information. For instance, models like GPT-4o (OpenAI, 2024) have achieved impressive results across numerous image-text benchmarks (Hudson & Manning, 2019; Lu et al., 2023; Yue et al., 2024), and have shown promise in real-world applications such as web navigation (Zheng et al., 2024a; He et al., 2024a). However, despite their strong performance in some vision-language applications, recent studies (Rahmanzadehgervi et al., 2024; Fu et al., 2024; Tong et al., 2024; Li et al., 2024c) reveal that state-of-the-art (SOTA) VLMs continue to struggle with fine-grained or low-level visual perception tasks that are trivial for humans, such as determining whether lines intersect or identifying the boundary of cars and roads. Addressing these limitations is crucial for enhancing VLMs' real-world applicability, as many practical scenarios—such as surgical robotics in healthcare or autonomous driving—require precise visual understanding beyond coarse-grained capabilities.

To address these challenges, prior works have explored a series of visual programming methods (Subramanian et al., 2023; Hu et al., 2024b; Gupta & Kembhavi, 2023; Surís et al., 2023; Mialon et al., 2023; Wu et al., 2023a; Yang et al., 2023c), where text queries are input into LLMs to generate code that invokes vision-specific models, using their outputs directly as predictions for the query. While these methods can harness the strengths of specialized vision models, their applicability is limited by the availability of predefined tools and cannot generalize to tasks that fall outside the

scope of existing solutions, making them far from a comprehensive visual perception framework. Another line of research focuses on prompting strategies to elicit foundation models' System-2 reasoning by involving iterative reasoning with intermediate tokens (Yu et al., 2024; Saha et al., 2024). A series of textual prompting methods (Wei et al., 2022; Saha et al., 2023; Yao et al., 2024; Besta et al., 2024) have been developed to generate structured reasoning steps, effectively eliciting System-2 reasoning for complex text-based tasks in large language models (LLMs). However, their effectiveness on fine-grained visual perception tasks for VLMs remains underexplored. Similarly, visual prompting methods (Lei et al., 2024; Yang et al., 2023a; Wu et al., 2024) guide VLMs in interpreting visual data by adding artifacts to images in various formats, such as bounding boxes, markers, or segmentation masks. While these methods have shown promise in some compositional visual reasoning tasks, it is still unclear whether VLMs can accurately perceive such visual prompts, let alone whether these techniques improve performance in fine-grained visual perception tasks.

To fill this gap, and inspired by recent advances in LLM-based agents (Wang et al., 2024d; Liu et al., 2023b; Significant-Gravitas, 2024; Wang et al., 2024a; Shen et al., 2024), we propose VIPACT (**VI**sual-**P**erception via VLM **A**gent **C**ollaboration and **T**ool-use), a general VLM-based framework that integrates multi-agent collaboration and vision expert models for fine-grained visual perception tasks. As illustrated in Figure 1, VIPACT consists of three core components: (1) an **orchestrator agent** that manages the workflow by analyzing tasks, coordinating agents, selecting tools, summarizing evidence, and deducing final answers; (2) **specialized agents** for tasks such as image captioning, visual prompt description, and image comparison, providing detailed visual analysis to the orchestrator; and (3) **vision expert models**, offering task-specific, fine-grained perceptual information to address VLMs' limitations. We empirically evaluate VIPACT against SOTA baselines across benchmarks that include diverse visual perception tasks—challenging for SOTA VLMs but easy for humans—featuring complex elements like visual prompts and multi-image inputs. VIPACT consistently outperforms previous baselines on all tasks, demonstrating its effectiveness and generalization. Additionally, our in-depth analysis highlights the importance of multi-agent collaboration in eliciting more detailed System-2 reasoning from VLMs, as well as the critical role of visual input for task planning, with improved error handling and evidence aggregation.

To summarize, our key contributions are as follows: (1) We introduce VIPACT, a novel multi-modal agent framework based on VLMs that synergizes multi-agent collaboration with vision expert models to enhance fine-grained visual perception. VIPACT is a fully autonomous system capable of handling a diverse range of visual perception tasks using a single prompt template. It leverages a VLM for task analysis, planning, and invoking multi-agent collaboration, with flexible plug-and-play modular components that allow for further extension. (2) We conduct extensive experiments across diverse visual perception benchmarks, demonstrating VIPACT's advantages over SOTA baselines; (3) We systematically analyze previous methods that have been proved to be effective in improving the general task-solving capabilities of foundation models for fine-grained visual perception, revealing their inconsistent effectiveness. (4) We present comprehensive ablation studies to assess the impact of multi-agent collaboration, visual input for task planning, and each component of VIPACT, along with a detailed error analysis identifying the limitations of current SOTA VLMs, which serve as bottlenecks for further improvement.

## 2 RELATED WORK

**VLM-based Agent.** As LLMs demonstrate increasing capabilities in task decomposition, instruction following, and structured output generation, LLM-based language agents have shown potential across a wide range of applications (Zhang et al., 2023c; Xi et al., 2023; Chen et al., 2023a; Significant-Gravitas, 2024; Shen et al., 2024; Deng et al., 2024a; Zhang et al., 2024e; Xie et al., 2024a; Liu et al., 2023b;a; Zhang et al., 2023a; Zhou et al., 2023). Recently, as the emergency of GPT-4o (OpenAI, 2024) with enhanced visual ability and low latency, VLMs have begun to be applied as agent backbones for vision-related tasks (Hu et al., 2024a). One prominent line of works focuses on Web Agents or GUI agents (Yan et al., 2023; Yang et al., 2023b; Zheng et al., 2024a; Xie et al., 2024c; Kapoor et al., 2024; Zhang et al., 2024a; Koh et al., 2024; Wang et al., 2024c; Lù et al., 2024; Zhang et al., 2024b; Deng et al., 2024b; You et al., 2024; Zheng et al., 2024b; Fan et al., 2024; Wang et al., 2024b; He et al., 2024b) which aim to interact with and navigate web interfaces and graphical user interfaces. Another line of works focuses on embodied agents designed to control robots (Nasiriany et al., 2024; Tan et al., 2024; Ma et al., 2024; Xie et al., 2024b; Yang et al.,

2024b; Szot et al., 2024), bridging the gap between language understanding and physical world interaction. Despite these advancements, to the best of our knowledge, there is no prior work focusing on building VLM-based agents specifically for natural image understanding or perception tasks. Eagle (Shi et al., 2024b) combines multiple vision encoders with a simple fusion strategy and a novel pre-alignment training stage to achieve SOTA performances on certain vision-language tasks. While Eagle also explores multi-expert collaboration in multimodal LLMs, its approach differs fundamentally from ours. Eagle focuses on integrating multiple internal vision encoders as "experts" within a single VLM. In contrast, `VipAct` leverages external pre-trained vision models as experts within a modular agent framework, interacting with but not modifying the architecture of the VLM.

**Visual Programming.** With the advancement of LLMs, particularly in code generation, recent work has begun utilizing LLMs as an interface for solving complex reasoning tasks with tools, using code generation as a proxy (Gao et al., 2023; Zhang et al., 2023c; 2024e;d; Schick et al., 2024). This approach has proven effective in reducing hallucinations in a wide range of tasks such as mathematical reasoning (Cobbe et al., 2021; Hendrycks et al., 2021). A line of research extends this concept to vision tasks (Subramanian et al., 2023; Hu et al., 2024b; Gupta & Kembhavi, 2023; Surís et al., 2023; Mialon et al., 2023; Wu et al., 2023a). MM-REACT (Yang et al., 2023c) integrates LLMs with various vision experts to perform multimodal reasoning tasks, following the prompt template of ReAct (Yao et al., 2023). ViperGPT (Surís et al., 2023) and VisProg (Gupta & Kembhavi, 2023) leverage LLMs to generate Python code that can be executed to perform visual reasoning tasks without additional training. However, these approaches typically use only the text query as input to the LLMs for code generation, neglecting the image input. Additionally, their workflows heavily depend on outputs from vision expert models, lack error-handling mechanisms, and involve tool selections that are to some extend hard-coded for specific, predefined tasks. These limitations restrict their effectiveness to simpler scenarios, such as question answering about main objects in images (Hudson & Manning, 2019; Suhr et al., 2019; Marino et al., 2019), without the capability for fine-grained visual perception or robust task generalization. Moreover, most existing methods lack specific designs for visual prompting within the image and are unable to handle tasks that require multiple images as input. This constrains their applicability to more complex visual reasoning scenarios that demand detailed perception and multi-image analysis. Table 1 provides a detailed comparison of the most closely related methods.

| Methods | Reas. | Tool | Multi-Ag. | Plan Img | Exec Img | Img Loop | Multi-Img | Vis. Prompt |
|---|---|---|---|---|---|---|---|---|
| ReAct (Yao et al., 2023) | ✓ | ✓ | ✗ | ✗ | ✗ | ✗ | ✗ | ✗ |
| MM-ReAct (Yang et al., 2023c) | ✓ | ✓ | ✗ | ✗ | ✓ | ✗ | ✗ | ✗ |
| ViperGPT (Surís et al., 2023) | ✗ | ✓ | ✗ | ✗ | ✓ | ✗ | ✗ | ✗ |
| VisProg (Gupta & Kembhavi, 2023) | ✗ | ✓ | ✗ | ✗ | ✓ | ✗ | ✗ | ✗ |
| CodeVQA (Subramanian et al., 2023) | ✗ | ✓ | ✗ | ✗ | ✓ | ✗ | ✗ | ✗ |
| **VIPACT (Ours)** | ✓ | ✓ | ✓ | ✓ | ✓ | ✓ | ✓ | ✓ |

Table 1: Comparison of VIPACT with other LLM/VLM-based agentic frameworks. ✓ indicates the presence of a specific feature in the corresponding framework, ✗ its absence. Column abbreviations: "Reas." for modules to elicit reasoning process, "Tool." for tool integration, "Multi-Ag." for multi-agent support, "Plan Img" for image input in planning, "Exec Img" for image input in execution, "Img Loop" for image use in iterative loops, "Multi-Img" for multi-image support, and "Vis. Prompt" for specific design for images containing visual prompts.

# 3 VIPACT FRAMEWORK

Our proposed framework, **VIPACT**, is illustrated in Figure 1. VIPACT consists of three main components: (1) **orchestrator agent** (Section 3.1), which controls the entire workflow by analyzing task requirements and task plans, initiating collaboration with other agents, selecting appropriate vision expert models, summarizing evidence from other agents or tools, and deducing the final answer. (2) **specialized agents** (Section 3.2), designed to handle specific tasks such as image captioning, visual prompt description, and image comparison. These agents provide detailed and relevant information to the orchestrator agent, facilitating the completion of complex visual perception tasks. (3) **vision expert models** (Section 3.3), which include specialized task-specific vision models that provide accurate, fine-grained perceptual information, addressing limitations of current VLMs. Intuitively, VIPACT enhances the VLM's System-2 reasoning by generating detailed intermediate reasoning

steps through multi-agent collaboration while leveraging the high-precision perceptual information from vision expert models.

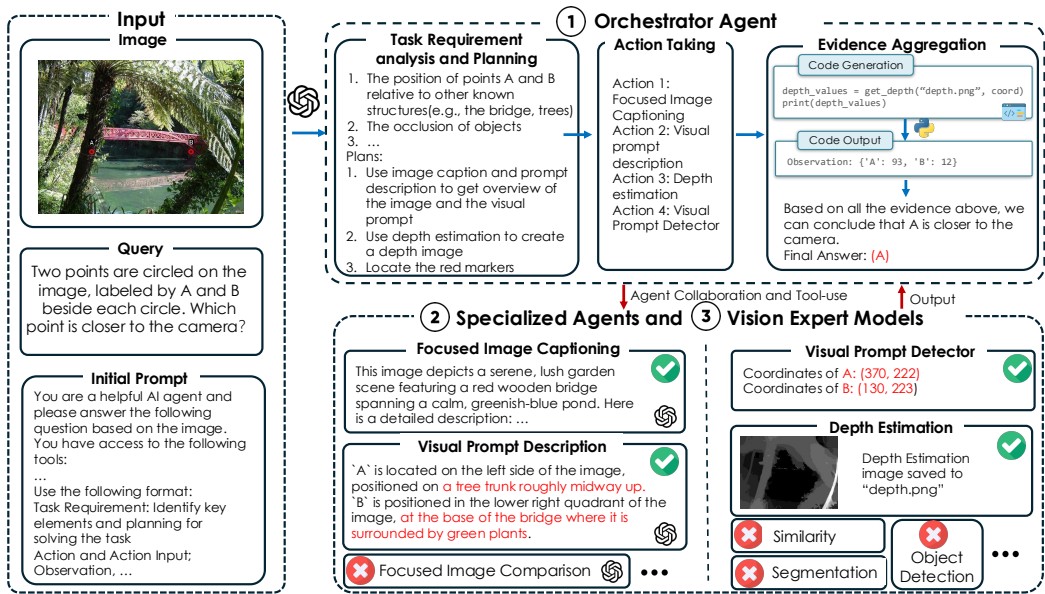

Figure 1: The VIPACT framework for visual perception. It consists of (1) an orchestrator agent for task analysis and coordination, (2) specialized agents for focused visual tasks, and (3) vision expert models for detailed visual analysis. The framework integrates both textual and visual outputs from the specialized agents and vision expert models to assist the orchestrator in addressing complex visual perception challenges. Note that not all agents and expert models are invoked in every instance—the orchestrator agent selectively activates the most relevant components based on the task characteristics and data. For complete task-solving processes of VIPACT, refer to the case studies in Appendix D.

### 3.1 ORCHESTRATOR AGENT

**Task Requirement Analysis and Planning:** Inspired by recent works (Yao et al., 2022; Huang et al., 2022; Yang et al., 2023c; Wang et al., 2023; Sun et al., 2024) that integrate reasoning, planning, and action in LLM-based agent frameworks, the orchestrator agent begins by analyzing the task requirements derived from the provided images and queries. This analysis identifies the key elements necessary to solve the problem and the corresponding critical visual features that must be acquired in subsequent steps of the agent's workflow, as well as other criteria derived from its own knowledge. The orchestrator agent then generates a detailed plan for tackling the task, outlining the concrete steps required to obtain the information needed to meet these requirements. For instance, in a depth estimation task as illustrated in Figure 1, the orchestrator agent would determine the essential requirements for comparing depth, such as identifying the specific objects targeted by the red circles and recognizing their relative positions to the camera.

**Tool Selection and Incorporation of Specialized Agents:** After analyzing the task requirements and formulating a plan, the orchestrator agent selects the appropriate tools and specialized agents to provide the visual information necessary to solve the task. Depending on the nature of the task, this may involve initiating collaboration with specialized agents designed for specific tasks or external vision expert models to gather comprehensive information. Details on these specialized agents and external vision expert models are provided in Sections 3.2 and 3.3.

**Evidence Summarization:** Once the tools and specialized agents have performed their respective tasks in separate environments, the orchestrator agent compiles and summarizes the collected evidence. This involves integrating the outputs from various tools and agents, ensuring that all relevant information is coherently synthesized to support the decision-making process. The orchestrator

agent also resolves conflicting evidence and double-checks the factuality of the information, as errors or hallucinations may arise from the expert models and specialized agents.

**Final Answer Deduction:** With the summarized evidence, the orchestrator agent deduces the final answer. It applies reasoning based on the accumulated information to arrive at a clear, unambiguous conclusion. Depending on the nature and format of the gathered data, the orchestrator agent may generate `Python` code, which is then executed by an external `Python` interpreter to derive the final answer. If the gathered information does not lead to a perfect answer, the orchestrator agent is designed to select the closest possible option based on the evidence, supplemented by its own perception and knowledge.

## 3.2 COLLABORATION WITH SPECIALIZED AGENTS

VIPACT incorporates three specialized agents to enhance its visual perception capabilities: focused image captioning, visual prompt description, and focused image comparison. These agents provide task-specific, detailed information to the orchestrator agent through function calling in a separate environment, integrating their outputs into the main reasoning process for comprehensive visual analysis. The three specialized agents used in our experiments are described below.

**Focused Image Captioning:** This agent generates detailed image descriptions, optionally emphasizing specific objects or elements relevant to the task by specifying a `focus` argument. The `focus` argument allows for targeted analysis, ranging from general descriptions to particular aspects like "a red car and the background buildings." This flexibility enables the orchestrator agent to obtain precise, task-relevant information from images. Empirical evidence demonstrates its effectiveness across a wide range of visual perception tasks, with the focus parameter providing fine-grained control over the generated descriptions.

**Visual Prompt Description:** Specializing in analyzing visual prompts within images (e.g., colored circles, bounding boxes, arrows, textual labels), this agent is crucial for interpreting visual annotations. It generates detailed descriptions of these elements, including their locations, characteristics, and most importantly, the regions or objects these visual prompts target. This enables the orchestrator agent to accurately interpret highlighted or annotated image sections. The agent has shown particular efficacy in tasks involving images with explicit visual prompts, significantly enhancing the system's ability to understand and reason about annotated visual data.

**Focused Image Comparison:** This agent analyzes multiple images, identifying similarities and differences with an optional focus on specific elements. Similarly, the `focus` parameter allows for targeted comparative analysis, either generally or on specific features as directed by the orchestrator agent. For example, this function can provide a detailed comparison of orientations of objects which can be useful in tasks such as multi-view reasoning. This capability is particularly valuable for tasks requiring multi-image input, such as change detection or pattern identification across images. Empirical results demonstrate this agent's exceptional effectiveness in tasks involving multiple image inputs, with the focus parameter enabling precise comparative analyses.

The complete prompts for these three specialized agents are in Appendix H. VIPACT uses these agents to break down complex visual tasks into manageable sub-tasks, with the orchestrator agent integrating their outputs for a comprehensive understanding. This modular approach ensures flexibility and precision, allowing for informed decisions and accurate responses. The architecture is also extensible, enabling easy integration of new agents to handle emerging visual tasks.

## 3.3 INTEGRATION OF VISION-EXPERT MODELS

VIPACT further enhances its visual perception capabilities by integrating a suite of vision-expert models, each specializing in specific aspects of image analysis. These models collaborate with the orchestrator agent through function calling, uniquely returning both textual data and processed images—making VIPACT among the earliest agent frameworks that incorporate **visual information directly into the reasoning workflow**. These vision-expert models provide fine-grained visual perception information that is often lacking in current VLM's pre-training data (Zhang et al., 2024c). The vision expert tools used in our experiments are described below.

- **Visual Prompt Detector:** Identifies and localizes annotated elements in images, such as circles, bounding boxes, or other highlighted regions. This tool is crucial for understanding visual instructions or annotations, enabling the agent to focus on relevant areas for analysis. It returns the coordinates of these visual prompts, which often serve as intermediate information to achieve the final answer.

- **Depth Estimator:** Analyzes spatial relationships within scenes, providing crucial information about the relative distances of objects from the camera. This tool enhances the agent's understanding of 3D structure in 2D images, vital for spatial reasoning tasks. It returns a grey-scale depth image that can be directly input into the orchestrator agent, allowing it to interpret depth information or combine it with other evidence to reach the final answer.

- **Object Detection:** Identifies and localizes objects within an image, providing the agent with a comprehensive inventory of visible objects, their locations, and sizes. This facilitates detailed scene understanding and object-centric reasoning. The tool returns both a processed image with detected objects' bounding boxes and textual information about these bounding boxes and objects.

- **Image Segmentation:** Offers precise delineation of image regions, separating objects, backgrounds, and distinct areas. This enables fine-grained analysis of image components, crucial for tasks requiring detailed understanding of object boundaries and spatial relationships. It returns images with segmentation masks along with corresponding textual information.

- **Embedding-based Similarity Computation:** Quantifies visual similarities across images or image regions by generating compact representations of visual content. This allows for nuanced comparisons and similarity assessments, particularly useful for tasks involving image retrieval or comparative analysis. It returns similarity scores based on the selected embedding model and specified similarity metrics, such as cosine similarity.

The complete function heads, including inputs, outputs, and descriptions for these vision expert models, are provided in the initial prompt for the orchestrator agents in Appendix H. This diverse toolkit empowers the orchestrator agent to dynamically select and deploy the most appropriate tools for each task, significantly enhancing the framework's ability to comprehend and reason about complex visual scenarios. The integration of processed images alongside textual outputs in the agent's workflow enables more nuanced and contextually rich visual reasoning. We provide an overview of the VipAct framework in Algorithm 1 with detailed explanations in Appendix G.

---

**Algorithm 1** VIPACT: **VI**sual-**P**erception via VLM **A**gent **C**ollaboration & **T**ool-use

---

**Require:** Set of visual inputs $\mathcal{V}$, a query $q$, a vision-language model $\mathcal{M}$, a set of tools $\mathcal{T} = \{T_1, \ldots, T_n\}$ including specialized agents and vision expert models, and the maximum iterations $K$

**Ensure:** An answer $a$ to the visual perception task

1: Initialize orchestrator agent $\mathcal{O}$ with $\mathcal{M}$ and $\mathcal{T}$
2: $\mathcal{P}_0 \leftarrow$ FORMATPROMPT$(\mathcal{V}, q)$      ▷ Format initial prompt with visual inputs and query
3: $t \leftarrow 0, \ \mathcal{S} \leftarrow \emptyset$      ▷ Initialize iteration counter and state
4: **while** $t < K$ and not ISTERMINATED$(\mathcal{S})$ **do**
5:    **if** $\exists T_i \in \mathcal{T} :$ ISREQUIRED$(T_i, \mathcal{S})$ **then**      ▷ Check if any tool is required
6:       $T^* \leftarrow \arg\max_{T_i \in \mathcal{T}}$ UTILITY$(T_i, \mathcal{S})$      ▷ Select most useful tool
7:       $\mathcal{O}_t \leftarrow$ EXECUTE$(T^*, \mathcal{S})$      ▷ Execute selected tool with the current state as input
8:       **if** CONTAINSVISUALDATA$(\mathcal{O}_t)$ **then**
9:          $\mathcal{V} \leftarrow \mathcal{V} \cup$ PROCESSVISUALDATA$(\mathcal{O}_t)$      ▷ Add new visual data if needed
10:   **else**
11:       $\mathcal{R}_t \leftarrow \mathcal{M}(\mathcal{P}_{t-1})$      ▷ Generate VLM output
12:       $\mathcal{O}_t \leftarrow$ INTERPRETOUTPUT$(\mathcal{R}_t)$      ▷ Interpret VLM output
13:    $\mathcal{P}_t \leftarrow$ UPDATEPROMPT$(\mathcal{P}_{t-1}, \mathcal{O}_t)$      ▷ Update prompt with new information
14:    $\mathcal{S} \leftarrow$ UPDATESTATE$(\mathcal{S}, \mathcal{O}_t); t \leftarrow t + 1$      ▷ Update state with new observations
15: $a \leftarrow$ EXTRACTANSWER$(\mathcal{S})$      ▷ Extract final answer from state
16: **return** $a$

---

| Method | Sim | Count | Depth | Jig | Fun.C | Sem.C | Spat | Local | Vis.C | Multi-v | Average |
|--------|-----|-------|-------|-----|-------|-------|------|-------|-------|---------|---------|
| *Text-based Prompting* | | | | | | | | | | | |
| Zero-shot | 65.44 | 50.83 | 64.52 | 60.00 | 57.69 | 56.83 | 79.92 | 56.00 | 86.05 | 60.15 | 63.74 |
| CoT | 63.70 | 65.00 | 73.39 | 62.00 | 57.69 | 57.55 | 82.52 | 60.66 | 82.56 | 53.38 | 65.85 |
| LtM | 62.22 | 64.17 | 70.97 | 62.67 | 55.38 | 55.40 | 76.22 | 59.02 | 83.14 | 45.86 | 63.51 |
| ToT | 64.44 | 58.33 | 71.70 | 64.00 | 57.69 | 59.71 | 83.22 | 61.48 | 78.49 | 50.38 | 64.94 |
| *Visual Prompting* | | | | | | | | | | | |
| SoM | 63.70 | 43.33 | 68.55 | 49.33 | 47.69 | 52.52 | 76.22 | 59.84 | 83.72 | 56.40 | 60.13 |
| *Mutli-modal Agent Framework* | | | | | | | | | | | |
| MM-ReAcT | - | 30.00 | 0.81 | - | - | - | 63.64 | 0.00 | - | - | - |
| ViperGPT | - | 29.17 | 0.00 | - | - | - | 48.95 | 18.85 | - | - | - |
| VisProg | - | 3.33 | 0.00 | - | - | - | 31.47 | 14.75 | - | - | - |
| **VIPACT (Ours)** | **81.48** | **70.00** | **90.80** | **68.00** | **61.50** | **60.40** | **86.70** | **63.11** | **91.28** | **62.63** | **73.79** |

Table 2: Results for visual reasoning tasks in Blink using GPT-4o. Note that "−" indicates methods that do not support multiple images. Our VIPACT consistently outperforms baselines on all tasks.

# 4 EXPERIMENT

**Setup.** Following previous works on web agents (Zheng et al., 2024a; He et al., 2024a; Liu et al., 2024a), we use GPT-4o (OpenAI, 2024) in our main experiment, which has proved to be the best model in visual agent benchmarks (Liu et al., 2024b). We also explore other VLMs in Appendix C and other implementation details can be found in Appendix A.

**Datasets.** To evaluate VLMs on visual perception tasks, we use two challenging datasets designed to test fine-grained visual perception. Dataset details are in Appendix B.

- **Blink** (Fu et al., 2024) includes diverse visual tasks solvable by humans "within a blink," yet difficult for SOTA VLMs. It features visual prompts such as bounding boxes and inter-leaved image-text formats, often with multiple images in a single query. We use Blink as the main benchmark to evaluate different methods.

- **MMVP** (Tong et al., 2024) is a benchmark for evaluating visual grounding in VLMs, using image pairs from "CLIP-blind pairs"—visually distinct images that are similar in CLIP embedding space. It focuses on nine basic visual patterns that are easy for humans but challenging for SOTA VLMs.

**Baselines.** We evaluate VIPACT against four categories of baselines: (1) Text-based prompting, including zero-shot instructional prompting, which inputs the image and question directly; chain-of-thought (CoT) prompting (Wei et al., 2022; Kojima et al., 2022), which appends "Let's think step-by-step" at the end of the instruction; Least-to-most prompting (LtM) (Zhou et al., 2022), which encourages LLMs to decompose the problem into more manageable sub-problems; and Tree-of-thought (ToT) prompting (Yao et al., 2024), which systematically explores multiple reasoning paths by maintaining a tree of intermediate steps. (2) Few-shot in-context learning Brown (2020), where in-context exemplars are selected using different strategies, including random selection, or selection based on the similarity of CLIP (Radford et al., 2021) or ViT Dosovitskiy et al. (2020) embeddings, which we analyze separately in Appendix E. (3) Visual Prompting, exemplified by Set-of-Mark (SoM) (Yang et al., 2023a), which overlays interpretable marks on semantically meaningful image regions, enhancing GPT-4V's fine-grained visual grounding on certain visual reasoning tasks. (4) Vision language agentic frameworks, including MM-ReAct (Yang et al., 2023c), which integrates LLMs with vision experts for multimodal reasoning and action through ReAct-style prompts (Yao et al., 2022); ViperGPT (Surís et al., 2023), which uses LLMs to generate Python code that composes vision and language models for visual reasoning tasks; VisProg (Gupta & Kembhavi, 2023), which generates visual programs from natural language instructions for complex tasks.

**Result Analysis.** Tables 2 and 3 present the performance of our proposed VIPACT framework and baseline methods on each sub-task of the Blink and MMVP datasets respectively. We make the following key observations: **(1) Text-based prompting methods do not consistently improve performance over zero-shot prompting.** Specifically, as shown in Tables 2 and 3, prior text-based prompting methods that have been effective in eliciting LLMs reasoning abilities — such as CoT

— can improve performance on some sub-tasks like visual similarity, object localization, counting, and spatial relations. However, for other tasks, the improvement is minimal or even negative. More advanced prompting techniques such as LtM and ToT exhibit similar phenomena. Empirically, we find that although these methods can elicit more detailed reasoning processes to reach the final answer, such reasoning steps are often not grounded in the visual elements of the image and can cause severe hallucinations. Therefore, we conclude that it is non-trivial to elicit VLMs' reasoning abilities for better general visual perception using text-based prompting methods that work for text-only LLMs. **(2) SoM can impair VLMs' fine-grained perception in most scenarios.** From the results on both datasets, we find that SoM adversely affects VLMs' performance on almost all tasks. Empirically, we observe that overlaying labeled masks can become cluttered when dealing with a large number of semantic objects or fine-grained object parts. These masks can negatively influence VLMs' perception of the original semantic objects and may confuse the models with the original visual prompts and their corresponding labels. Consequently, we conclude that although SoM demonstrates effectiveness in some compositional reasoning tasks with a limited number of semantic objects, it does not generalize well to a broader range of visual perception tasks, especially those requiring visual prompt understanding. **(3) Previous visual programming methods exhibit poor generalization ability.** As shown in the results, these methods perform adequately only on a limited number of tasks such as spatial relations and counting, which are similar to those in commonly used visual question answering (VQA) datasets (Hudson & Manning, 2019; Suhr et al., 2019; Marino et al., 2019). Upon examining their reasoning processes and generated code, we find that the code can only call a limited set of tools predefined in the initial prompt, lacking additional logic to handle scenarios where their predefined tools are unsupported or when errors occur. Another limitation is their inability to support images with visual prompts, preventing them from locating visual prompts and proceeding with subsequent operations. For example, in tasks like depth estimation, their performance is close to zero because they cannot locate the red circles, resulting in non-executable generated code with no schema to handle such incapability. Moreover, since the code in these methods is generated solely based on the text query without considering the image, it lacks the flexibility to adapt to different image characteristics. These observations highlight the need for designing a generalizable agent framework that can leverage both vision expert models and the inherent flexibility of VLMs themselves. **(4) VIPACT consistently achieves the best performance across all sub-tasks in Blink and MMVP, demonstrating its effectiveness and generalization ability.** By thoroughly examining VIPACT's reasoning steps, we observe that, compared to text-based and visual prompting methods, VIPACT can effectively invoke specialized agents or vision expert models to enhance its understanding of the image. Moreover, VIPACT does not solely rely on the outputs from these agents, as the evidence they provide may be incorrect or errors may occur. Instead, it aggregates all useful evidence with additional reasoning steps to infer the final answer, showcasing its ability to handle uncertainties and integrate multiple sources of information which ensures its superior generalization ability. Figure 3 and 4 in Appendix D showcase the complete reasoning process of VIPACT to solve visual perception tasks.

## 5 ABLATION STUDY

To evaluate the effectiveness of various components in our VIPACT framework, we further conduct a series of ablation studies. These studies involve removing or modifying key components of the VIPACT framework to assess their impact on performance across different visual reasoning tasks. The ablation studies are as follows: **(1) Removal of multi-agent collaboration**: We removed the specialized agents and incorporated their prompts as instructions directly into the orchestrator agent to evaluate the importance of multi-agent collaboration. **(2) Removal of image input for orchestrator agent:** We modified the input to the orchestrator agent to only include image paths as text, rather than the actual images which means the image is not visible to the orchestrator agent

| Method | Accuracy (%) |
|---|---|
| Zero-shot | 68.0 |
| CoT | 61.0 |
| LtM | 66.0 |
| ToT | 66.0 |
| SoM | 62.0 |
| MM-ReAcT | 6.67 |
| ViperGPT | 53.0 |
| VisPro | 39.0 |
| **VIPACT (Ours)** | **70.7** |

Table 3: Results of different methods using GPT-4o on MMVP.

but still can be served as input for other specialized agents or vision expert models. This setup follows the paradigm used in previous works (Surís et al., 2023; Gupta & Kembhavi, 2023) and tests

| Method | Sim | Count | Depth | Jig | Fun.C | Sem.C | Spat | Local | Vis.C | Multi-v |
|---|---|---|---|---|---|---|---|---|---|---|
| *Variants of* VIPACT | | | | | | | | | | |
| VIPACT (Full) | **81.48** | **70.00** | **90.80** | **68.00** | **61.50** | **60.40** | **86.70** | 63.11 | **91.28** | **62.63** |
|   w/o Multi-agent | 80.00 | 67.50 | 75.00 | 66.00 | 58.46 | 59.71 | 82.52 | **63.93** | 85.47 | 48.87 |
|   w/o Visual Input | 77.78 | 59.71 | 69.35 | 61.33 | 53.85 | 51.08 | 83.22 | 60.66 | 78.49 | 48.12 |
|   w/o Spec. Agents | 65.72 | 62.45 | 85.62 | 62.32 | 55.25 | 56.32 | 81.96 | 58.49 | 75.48 | 46.75 |
|   w/o Vision Expert | 64.34 | 57.44 | 72.58 | 65.67 | 59.42 | 58.59 | 81.37 | 57.44 | 83.63 | 56.40 |

Table 4: Ablation study results of VIPACT on the Blink benchmark. VIPACT (Full) represents the complete framework with all components, while the other variants exclude specific components.

the effectiveness of direct visual input to the orchestrator agent. **(3) Removal of specialized agents:** We removed all specialized agents to assess their collective impact on the VIPACT's performance. **(4) Removal of vision expert models:** We eliminated all vision-expert models to evaluate their contribution to VIPACT's capabilities.

The results of these ablation studies are presented in Table 4 and 5. From these results, we derive the following key insights: **(1) Multi-agent collaboration enhances detailed reasoning** : The removal of multi-agent collaboration led to a consistent performance decline across nearly all tasks. By comparing the reasoning steps between the complete **VIPACT** and this ablated version, we observed that, although the orchestrator agent had the same instructions, multi-agent collaboration enabled the generation of a much more detailed analysis of the images

| Method | Accuracy (%) |
|---|---|
| VIPACT | **70.7** |
|   w/o Multi-agent | 68.0 |
|   w/o Visual Input | 54.0 |
|   w/o Spec. Agents | 67.0 |
|   w/o Vision Expert | 66.0 |

Table 5: Ablation study results of VIPACT on the MMVP benchmark.

(over 80% more generated tokens [1]), such as thorough image captioning. This phenomenon aligns with observations in LLMs (Wu et al., 2023b; Hong et al., 2023; Qian et al., 2023; Park et al., 2023; Liu et al., 2023b), where collaboration among multiple agents enhances the ability to solve complex tasks by providing comprehensive reasoning from diverse perspectives. **(2) Direct image input to the orchestrator agent is essential for flexible task planning and error handling**: As demonstrated in Table 4 and 5, removing the image input to the orchestrator agent significantly impairs performance on both datasets. By examining the reasoning process, we observe that without direct visual input, the orchestrator agent's task requirement analysis and planning become more general and less specific to individual data points, negatively affecting subsequent tool usage—particularly in setting input parameters (e.g., the `focus` parameters for specialized agents). Furthermore, the orchestrator agent struggles to effectively aggregate conflicting evidence or handle error messages from different tools without its own understanding of the image. **(3) Specialized agents and vision expert models significantly contribute to performance**: Although specialized agents are also VLMs, they focus intently on analyzing specific aspects of the image's visual information (e.g., prompt description) without being distracted by other instructions such as format requirements or output structures. Prior work has also shown that such distractions can hinder the reasoning process of LLMs (Tam et al., 2024). Vision expert models, on the other hand, can perform pixel-level analyses that even SOTA VLMs can not handle well, effectively aiding the orchestrator agent in achieving the correct answer. As demonstrated in Table 4 and 5, removing these components leads to a noticeable decline in performance, underscoring their importance within the framework. Overall, our VIPACT framework combines the flexibility and planning of VLMs with the precision of vision expert models, creating a cohesive system where each component is essential to performance.

## 6 ERROR ANALYSIS

To thoroughly examine the limitations of GPT-4o's visual perception capabilities and to better understand the challenges faced by SOTA VLMs as well as the bottlenecks of our VIPACT framework, we conducted a detailed error analysis. Following the practices established by prior works (Zhou et al., 2022; Chen et al., 2023b; Zhang et al., 2024d), we randomly sampled 20 error cases from

---

[1]The number of tokens in the reasoning steps for both methods remains well below the token limit.

each sub-task within the Blink and MMVP datasets. The errors were categorized as follows, with corresponding percentages:

- **Failure to perceive small object parts (17%):** The model often overlooks small, semantically important components of objects, which are crucial for precise visual understanding.
- **Difficulty distinguishing closely positioned visual prompts (15%):** The model struggles to differentiate visual prompts that are spatially proximate, leading to confusion between their targeted regions.
- **Challenges in fine-grained spatial reasoning (24%):** Tasks requiring high spatial resolution, such as boundary recognition, highlight the model's **bias towards foreground objects over backgrounds**. For instance, in cases where a red circle is meant to highlight a point in the sky near a car, the model frequently misinterprets the circle as being associated with the car, rather than the sky.
- **Misinterpretation of relative object positions (14%):** A significant source of error arises when the spatial arrangement of objects differs from real-world expectations. The model often lacks the ability to infer spatial relations from the objects' perspectives, focusing instead on the camera's viewpoint.
- **Failure to recognize object orientation (13%):** The model encounters difficulty in discerning object orientation, which leads to errors in recognizing object parts. For example, in images of bicycles, the model struggles to distinguish between the left and right pedals based on their spatial orientation.
- **Miscellaneous errors (17%):** This category includes various other issues, such as failure to detect subtle color differences, inaccuracies in multi-image fine-grained structure cooresponding, and instances where the model either refuses to respond or misinterprets instructions.

Case studies illustrating these errors are available in Appendix D. Our analysis denotes that while VIPACT demonstrates significant improvements in VLM visual perception, fine-grained perception remains a bottleneck for further improvement. Specifically, the model lacks the **spatial intelligence or imaginative abilities** (Chen et al., 2018; Huang et al., 2024) necessary to infer the relative positions of objects, not just based on their pixel positions in the image (from the camera's perspective projection) but in the context of real-life scenes. Noticeably, these limitations hinder the model's ability to accurately interpret visual prompts and process tasks involving multiple image inputs. We also examine the significance of multiple image inputs for VLMs in Appendix F.

## 7 CONCLUSION

We introduce **VIPACT**, a VLM-based agent framework that synergizes multi-agent collaboration and vision expert models for fine-grained visual perception tasks. By combining the planning and function-calling capabilities of SOTA VLMs, VIPACT enhances VLMs' System-2 reasoning through multi-agent interactions and integrates high-precision, pixel-level information from specialized vision models. Our comprehensive experiments across a diverse range of visual perception tasks demonstrate that VIPACT achieves SOTA performance, outperforming previous baselines. The comprehensive ablation study highlights the critical role of multi-agent collaboration in eliciting detailed information for reasoning, as well as the importance of image input in task planning. Furthermore, our error analysis highlights several inherent limitations in current SOTA VLMs that form bottlenecks in our framework, offering valuable insights for future improvements.

Our work has several limitations: (1) The inference cost of VLMs can be high, as our framework often requires multiple inferences, including tool calls and specialized agents' outputs, increasing computational overhead. This is a common issue across all multi-agent frameworks that involve complex reasoning steps, and it is inevitable when generating more detailed reasoning. (2) VIPACT relies heavily on GPT-4o due to its superior instruction-following and function-calling abilities for our needs. While we have explored other VLMs, such as LLaVa-OneVision-7B (Li et al., 2024a) in Appendix C, they struggle with following instructions such as formatting requirements. However, VIPACT is a general framework and can be adapted to other VLMs as they evolve. (3) We did not design task-specific vision expert tools for every task, but VIPACT's modular architecture allows easy integration of additional tools and agents in a plug-and-play manner.

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

# A  IMPLEMENTATION DETAILS

For main experiments, we use the `gpt-4o-2024-05-13` model from Azure OpenAI API. Following previous works (Fu et al., 2024) to ensure reproducibility, we set the temperature to 0 for all VLM inference and set the maximum number of tokens to 2048. For components of VIPACT, we use the same `gpt-4o-2024-05-13` model for the implementation of orchestrator agents and specialized agents. For the implementation of vision expert models, we use the `Depth-Anything-V2-Small-hf` checkpoint (Yang et al., 2024a) for depth estimation, the Segment Anything Model (SAM) (Kirillov et al., 2023) for segmentation, the YOLOv8 model (Hussain, 2023) from Ultralytics for object detection, and the `clip-vit-base-patch32` (Radford et al., 2021) for similarity comparison using cosine similarity. For experiments with LLaVA, we use the latest SOTA `llava-onevision-qwen2-7b-ov` (Li et al., 2024a), which is one of the few VLMs that support multiple images as inputs and achieves SOTA results on various vision-language benchmarks (Li et al., 2024b; Bansal et al., 2020) compared to other open-source models of similar size. For the implementation of all prompting baselines, we adopt the codebase from the original Blink and MMVP papers and use the exact same settings, including the method for computing performance. For the implementation of baselines MM-ReAct, ViperGPT, and VisProg, we adopt the original codebase they provide, except that the backbone model is replaced with GPT-4o, as their original models such as Codex (Chen et al., 2021) are not available and to ensure fair comparison. For the implementation of few-shot in-context learning, the embedding models' checkpoints we use are `clip-vit-base-patch32` and `vit-base-patch16-224` (Alexey, 2020). For all experiments, we run three times and report the average number. For the results in Table 2 and 3, we conduct significance tests following Berg-Kirkpatrick et al. (2012). The average estimate of p-value is 0.006 ($< 0.01$) between VIPACT and SOTA baselines, demonstrating significant differences. The total inference time for our VIPACT on Blink and MMVP is less than 2 hours, which is acceptable. Our computational resources consist of a Linux server with 4 NVIDIA A100-40G GPUs.

# B  DATASET DETAILS

In this section, we provide the details of the dataset used in our experiments. The Blink (Fu et al., 2024) dataset contains a variety of tasks that evaluate different aspects of VLMs' perception capabilities. In our paper, we specifically focus on the following sub-tasks: Similarity (**Sim**), Counting (**Count**), Depth Estimation (**Depth**), Jigsaw Puzzle (**Jig**), Functional Correspondence (**Fun.C**), Semantic Correspondence (**Sem.C**), Spatial relation (**Spat**), Local Correspondence (**Local**), Visual Correspondence (**Vis.C**), and Multi-view Reasoning (**Multi-v**). The dataset is divided into validation and test sets, with the number of data points for each sub-task as shown in Table 6.

| Sub-task | Sim | Count | Depth | Jig | Fun.C | Sem.C | Spat | Local | Vis.C | Multi-v |
|---|---|---|---|---|---|---|---|---|---|---|
| **Validation** | 135 | 120 | 124 | 150 | 130 | 139 | 143 | 122 | 172 | 133 |
| **Test** | 136 | 120 | 124 | 150 | 130 | 140 | 143 | 125 | 172 | 133 |

Table 6: Number of data points for each sub-task in the validation and test sets of Blink.

The tasks and the corresponding datasets are described in the original Blink paper. Each sub-task is designed to challenge different aspects of the model's perceptual reasoning capabilities, as detailed in the main text of our paper. Following previous works (Hu et al., 2024c), we exclude datasets focused on compositional reasoning like IQ testing or commonsense reasoning, as they do not directly assess visual perception and more focus on compositional reasoning.

Another dataset we use in this work is the Multimodal Visual Patterns (MMVP) dataset (Tong et al., 2024) which consists of 150 CLIP-blind image pairs and 300 associated visual questions, designed to probe nine core visual patterns: orientation, presence of specific features, state, quantity, positional context, color, structure, text, and viewpoint. Human participants achieved 95.7% accuracy, while state-of-the-art MLLMs, including GPT-4V and Gemini, performed significantly worse. The dataset highlights fundamental failures in visual grounding tasks and serves as a benchmark for advancing VLMs' visual perception ability.

| Method | Sim | Count | Depth | Jig | Fun.C | Sem.C | Spat | Local | Vis.C | Multi-v | Overall |
|--------|-----|-------|-------|-----|-------|-------|------|-------|-------|---------|---------|
| *Text-based Prompting* | | | | | | | | | | | |
| Zero-shot | 78.52 | 60.83 | 70.97 | 72.67 | 44.62 | 51.08 | 74.13 | 57.38 | 81.98 | 55.64 | 64.78 |
| CoT | 81.48 | **64.17** | 78.23 | 68.67 | 42.31 | 53.96 | 78.32 | 60.66 | 81.98 | 51.88 | 66.17 |
| LtM | 84.44 | 62.50 | 75.81 | 73.33 | 46.92 | 50.36 | 73.43 | 63.11 | 84.30 | 54.14 | 66.83 |
| ToT | 82.23 | 61.52 | 74.84 | 68.96 | 45.42 | 52.61 | 75.34 | 61.25 | 82.21 | 52.86 | 65.72 |
| *Visual Prompting* | | | | | | | | | | | |
| SoM | 60.74 | 55.00 | 65.32 | 62.00 | 47.69 | 43.88 | 74.13 | 59.02 | 74.42 | 53.38 | 59.56 |
| *Multi-modal Agent Framework* | | | | | | | | | | | |
| VipAct | **84.44** | **64.17** | **89.42** | **74.00** | **48.74** | **57.55** | **79.57** | **70.48** | **86.05** | **59.42** | **71.38** |

Table 7: Results for visual reasoning tasks in Blink using Gemini-1.5-Pro. Our `VipAct` consistently outperforms baselines on almost all tasks.

## C    EXPLORATION OF DIFFERENT VLMS

In addition to the GPT-4o used in our main experiments, we also evaluate other VLMs on our tasks. Specifically, we explore five additional SOTA VLMs, including (1) open-source models: LLaVA-OneVision-7B (Li et al., 2024a), the latest open-source model in the LLaVA series, InternVL-2-Pro (Chen et al., 2023d; 2024a), and Llama-3.2-90b-vision (Dubey et al., 2024); and (2) close-source models: Gemini-1.5-Pro (Team et al., 2024) and Claude-3.5-Sonnet (Anthropic, 2024).

For open-source models, we find that applying VIPACT's prompt (described in Section H) reveals significant limitations. These VLMs often fail to follow key instructions, such as adhering to the required format, which is critical for extracting the tool-use indicators necessary for integrating external tools. Furthermore, they frequently generate image captions even when no such instruction is provided, suggesting a bias towards image captioning or description tasks.

To evaluate these open-source VLMs comprehensively, we apply prompting baselines and report the results on the Blink benchmark and MMVP in Table 10 and Table 9. These results demonstrate that while LLaVA-OneVision-7B achieves above-random accuracy on tasks like object counting and spatial relations—typical of standard VQA problems found in prior datasets (Li et al., 2024b; Bansal et al., 2020)—it performs near or below random on other tasks. We also observe significant positional biases (Zhang et al., 2024f; Shi et al., 2024a), with this model frequently predicting the first option for most data points within a task. In contrast, InternVL-2-Pro and Llama-3.2-90b-vision exhibit better performance, though still significantly behind GPT-4o. These findings indicate that current open-source SOTA VLMs struggle with generalizing to more complex or non-standard VQA tasks, lacking the fine-grained perception capabilities necessary for broader applicability. Moreover, alternative prompting strategies do not yield noticeable improvements over the zero-shot baseline for these models.

In contrast, the two additional close-source VLMs—Gemini-1.5-Pro and Claude-3.5-Sonnet—demonstrate instruction-following abilities comparable to GPT-4o, allowing effective application of our VIPACT framework. As shown in Tables 7 and 8, applying VIPACT on these models consistently outperforms previous prompting baselines, achieving significant improvements. These results highlight the effectiveness and generalization capability of our approach when used with models possessing strong instruction-following capabilities.

## D    CASE STUDIES

To intuitively demonstrate the effectiveness of our proposed VIPACT and highlight the bottlenecks of current SOTA VLMs, we present a series of case studies showcasing both failure (Figure 2) and success cases (Figures 3 and 4) of our method.

In Figure 2, we observe instances where VLM-based specialized agents in VIPACT make reasoning errors, as categorized in Section 6. Although VIPACT includes an error-handling mechanism to reassess the evidence, these errors can still mislead the orchestrator agent, leading to incorrect inferences. For instance, in the top case of Figure 2, the VLM fails to accurately infer the orientation of the bicycle in the left image, mistakenly identifying the left pedal as the reference point based on the camera's perspective. In the middle case, the VLM overlooks the small portion of the cap's

| Method | Sim | Count | Depth | Jig | Fun.C | Sem.C | Spat | Local | Vis.C | Multi-v | Overall |
|---|---|---|---|---|---|---|---|---|---|---|---|
| *Text-based Prompting* | | | | | | | | | | | |
| Zero-shot | 85.19 | 67.50 | 66.13 | 58.00 | 58.00 | 44.60 | 72.03 | 57.38 | 73.84 | 48.12 | 63.08 |
| CoT | 86.72 | **68.33** | 71.77 | 61.33 | 52.31 | 41.73 | 77.62 | 50.00 | 81.98 | 44.36 | 63.62 |
| LtM | 87.42 | 67.42 | 68.42 | 59.97 | 58.00 | 45.13 | 73.82 | 57.47 | 74.29 | 47.86 | 63.98 |
| ToT | 86.90 | 67.53 | 69.48 | 57.35 | 59.46 | 43.72 | 74.92 | 58.49 | 76.14 | 46.38 | 64.04 |
| *Visual Prompting* | | | | | | | | | | | |
| SoM | 82.65 | 62.78 | 63.81 | 56.79 | 56.73 | 39.58 | 72.00 | 52.47 | 73.74 | 44.63 | 60.52 |
| *Multi-modal Agent Framework* | | | | | | | | | | | |
| VipAct | **88.89** | 67.96 | **88.59** | **65.33** | **60.42** | **50.13** | 78.82 | **61.54** | **83.72** | **49.57** | **69.50** |

Table 8: Results for visual reasoning tasks in Blink using Claude-3.5-Sonnet. Our `VipAct` consistently outperforms baselines on almost all tasks.

| Method | LLaVA-OneVision-7B | InternVL-2-Pro | Llama-3.2-90b-vision |
|---|---|---|---|
| Random | 25.00 | 25.00 | 25.00 |
| Zero-shot | 29.67 | 60.00 | 57.33 |
| CoT | 30.33 | 57.33 | 59.33 |
| LtM | 30.00 | 58.67 | 57.33 |
| ToT | 31.33 | 60.00 | 59.38 |
| SoM | 27.00 | 45.33 | 51.33 |

Table 9: Results of different open-source VLMs with different prompting methods on the MMVP benchmark, including a random baseline for comparison.

| Method | Sim | Count | Depth | Jig | Fun.C | Sem.C | Spat | Local | Vis.C | Multi-v |
|---|---|---|---|---|---|---|---|---|---|---|
| Random | 50.00 | 25.00 | 50.00 | 50.00 | 25.00 | 25.00 | 50.00 | 50.00 | 25.00 | 50.00 |
| Zero-shot | 47.41 | 63.33 | 51.61 | 52.67 | 20.00 | 23.02 | 72.73 | 50.82 | 23.26 | 44.36 |
| CoT | 44.44 | 57.20 | 54.03 | 52.67 | 20.77 | 25.90 | 76.22 | 43.44 | 22.67 | 35.34 |
| LtM | 45.93 | 56.67 | 51.61 | 52.67 | 15.38 | 28.87 | 72.03 | 50.82 | 30.81 | 42.11 |
| ToT | 47.41 | 63.33 | 50.00 | 52.67 | 15.38 | 24.46 | 72.03 | 50.82 | 23.26 | 44.36 |
| SoM | 47.41 | 46.67 | 54.03 | 52.67 | 23.85 | 21.58 | 72.73 | 41.80 | 19.19 | 31.58 |

Table 10: Result of baseline methods evaluated using LLaVa-OneVision-7B on the Blink dataset.

brim, leading to an incorrect prediction. Finally, the bottom case demonstrates how the camera's perspective makes it appear as though the apples are positioned above the orange when in reality, they are on the same plate at the same height. These examples highlight the limitations in visual intelligence exhibited by SOTA VLMs such as GPT-4o, particularly in tasks requiring fine-grained spatial reasoning.

In Figures 3 and 4, we present two examples that demonstrate the complete reasoning process of our VIPACT, integrating vision expert models and specialized agents. Figure 3 illustrates a scenario where the orchestrator agent sequentially invokes vision expert models, including a Visual Prompt Detector and a Depth Estimator, to accurately determine the depth values of two red points in the image, ultimately arriving at the correct answer. In contrast, we observe that GPT-4o is unable to perceive such depth information on its own. Figure 4 presents a case where no existing vision tools can directly solve the problem. Here, the orchestrator agent introduces a specialized agent specifically designed for visual prompt description. This agent provides a detailed analysis of each visual prompt (marked by red circles) in the second image, leading to the correct prediction. These two examples effectively illustrate the strength of our VIPACT framework in integrating vision expert models and specialized agents to enhance reasoning capabilities.

| Dataset | Model | |
| --- | --- | --- |
| | GPT-4o | LLaVA-OneVision-7B |
| Sim | 59.51 (↓ -5.93) | 45.93 (↓ -1.48) |
| Jig | 57.78 (↓ -2.22) | 52.67 (→ 0.00) |
| Fun.C | 53.34 (↓ -4.35) | 20.00 (→ 0.00) |
| Sem.C | 56.60 (↓ -0.23) | 24.46 (↑ +1.44) |
| Vis.C | 83.91 (↓ -2.14) | 18.60 (↓ -4.66) |
| Multi-v | 51.38 (↓ -8.77) | 29.32 (↓ -15.04) |
| **Overall** | 60.42 (↓ -3.94) | 31.83 (↓ -3.29) |

Table 11: Results of GPT-4o and LLaVA-OneVision-7B on Blink tasks requiring multiple image inputs, where multiple images are concatenated into a single image during inference. Performance changes compared to the zero-shot baseline with multiple image inputs are indicated in parentheses.

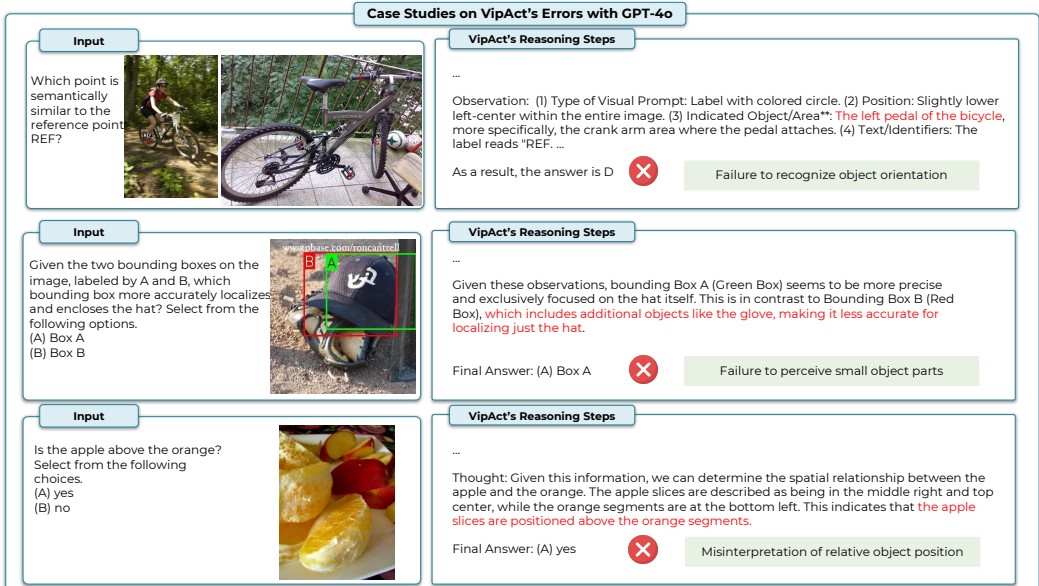

Figure 2: Types of error cases in VIPACT with their corresponding reasoning steps.

# E    FEW-SHOT IN-CONTEXT LEARNING

In this section, we examine the effectiveness of few-shot in-context learning in visual perception tasks using various VLMs, including GPT-4o and LLaVA-OneVision-7B. Following previous works (Brown, 2020; Alayrac et al., 2022; Awadalla et al., 2023; Zhao et al., 2023; Jiang et al., 2024), we append a series of (image(s), question, answer) triplets—ranging from 1 to 5—before the test query, within the overall instruction. This setup has been shown to enhance performance in LLMs on a wide range of NLP tasks. Additionally, prior research indicates that LLMs can be sensitive to the selection of in-context exemplars (Nguyen & Wong, 2023; Zhang et al., 2022; Agrawal et al., 2023; Chen et al., 2023c; Zhang et al., 2023b). To explore this, we employ three different strategies for exemplar selection: (1) Randomly select a specified number of exemplars. (2) Select exemplars based on top-K similarity using the averaged CLIP embedding of images, which captures both textual semantics and visual information (Radford et al., 2021). (3) Select exemplars based on top-K similarity using ViT embeddings (Alexey, 2020), which focus purely on visual features.

Table 12 presents the results of few-shot in-context learning with GPT-4o on the Blink benchmark. We observe that for certain tasks, such as object counting and spatial relations, few-shot learning significantly decreases performance compared to other baselines (see Table 2). However, for tasks like visual correspondence, few-shot in-context learning yields competitive results. Interestingly, as the number of shots increases, no consistent performance trend emerges across the different retrieval

| Method (# of shots) | Sim | Count | Depth | Jig | Fun.C | Sem.C | Spat | Local | Vis.C | Multi-v |
|---|---|---|---|---|---|---|---|---|---|---|
| *Randomly Choose One From the Options* | | | | | | | | | | |
| Random | 50.00 | 25.00 | 50.00 | 50.00 | 25.00 | 25.00 | 50.00 | 50.00 | 25.00 | 50.00 |
| *Randomly Select In-context Exemplars* | | | | | | | | | | |
| 1-shot | 65.93 | 25.00 | 71.77 | 64.00 | 60.77 | 56.12 | 45.45 | 61.48 | 86.05 | 48.12 |
| 2-shot | 42.22 | 25.83 | 73.39 | 62.00 | 58.46 | 58.99 | 47.55 | 58.20 | 88.37 | 55.64 |
| 3-shot | 52.59 | 26.67 | 51.61 | 64.00 | 57.69 | 57.55 | 47.55 | 60.66 | 88.37 | 45.11 |
| 4-shot | 64.44 | 21.67 | 66.13 | 61.33 | 54.62 | 55.40 | 46.85 | 61.48 | 88.37 | 50.38 |
| 5-shot | 56.30 | 30.00 | 70.16 | 61.33 | 60.77 | 59.71 | 49.65 | 59.84 | 87.79 | 53.38 |
| *Select In-context Exemplars Based on CLIP Embedding Similarity* | | | | | | | | | | |
| 1-shot | 78.52 | 20.00 | 66.13 | 52.00 | 56.15 | 56.12 | 44.06 | 58.20 | 87.79 | 51.88 |
| 2-shot | 60.00 | 30.00 | 61.29 | 60.67 | 54.62 | 53.96 | 47.55 | 63.11 | 84.88 | 51.88 |
| 3-shot | 52.59 | 26.67 | 59.68 | 66.00 | 59.23 | 54.68 | 46.15 | 61.48 | 89.53 | 51.13 |
| 4-shot | 57.04 | 31.67 | 68.55 | 66.00 | 55.38 | 56.12 | 45.45 | 63.11 | 88.95 | 56.40 |
| 5-shot | 60.00 | 25.00 | 64.52 | 62.67 | 58.46 | 53.24 | 47.55 | 59.84 | 87.21 | 54.89 |
| *Select In-context Exemplars Based on VIT Embedding Similarity* | | | | | | | | | | |
| 1-shot | 73.33 | 21.67 | 66.94 | 55.33 | 56.15 | 49.64 | 46.85 | 56.56 | 91.28 | 48.87 |
| 2-shot | 63.70 | 28.33 | 62.10 | 60.00 | 57.69 | 51.80 | 47.55 | 63.93 | 88.37 | 52.63 |
| 3-shot | 57.78 | 27.50 | 62.90 | 64.67 | 57.69 | 53.24 | 46.85 | 60.66 | 89.53 | 50.38 |
| 4-shot | 46.67 | 30.83 | 61.29 | 64.67 | 56.92 | 53.24 | 48.25 | 59.02 | 89.53 | 48.87 |
| 5-shot | 54.07 | 30.00 | 66.13 | 68.00 | 60.77 | 51.08 | 45.45 | 61.40 | 87.79 | 51.13 |

Table 12: Few-shot in-context learning results on the Blink dataset using GPT-4o, evaluated with varying numbers of exemplars and three retrieval methods.

methods. Moreover, we do not observe significant or consistent performance differences between the retrieval strategies.

Table 13 shows the results of few-shot in-context learning with LLaVA-OneVision-7B on Blink. Here, we find that performance on almost all sub-tasks is not significantly better than random guessing, even for tasks like object counting and spatial relations, where this model performs much better in baseline settings. Further examination of the outputs reveals that the positional biases identified in Section C persist and even worsen with few-shot prompting, as the model tends to predict the first option in most cases.

In conclusion, while few-shot in-context learning can be effective for some visual perception tasks with GPT-4o, it does not consistently outperform zero-shot baselines and can sometimes negatively impact performance. Additionally, retrieval strategies based on different embedding spaces do not show a clear advantage. For the open-source VLM LLaVA-OneVision-7B, few-shot in-context learning offers no noticeable benefits on these tasks and may even amplify existing biases, further degrading performance.

# F EXPLORING THE IMPORTANCE OF MULTIPLE IMAGE INPUTS TO VLMS

Understanding the relationships between multiple images is crucial for certain visual perception tasks and real-world applications. However, only a few closed-source VLMs (Reid et al., 2024) and a very limited number of open-source VLMs natively support multiple image inputs. For models that do not support this feature, the common practice is to concatenate multiple images into a single image with added margins and input this combined image into the VLM. To investigate this problem, we conduct experiments using concatenated images for tasks requiring multiple image inputs, utilizing both GPT-4o and LLaVA-OneVision-7B. As shown in Table 11, we observe a noticeable decline in performance for both models when multiple images are concatenated into a single image. This decline is particularly consistent with GPT-4o, indicating that concatenating images introduces challenges that these VLMs struggle to handle effectively. This suggests that native support for multiple image inputs is important for maintaining performance, and concatenating images is not the ideal practice for VLMs.

| Method (# of shots) | Sim | Count | Depth | Jig | Fun.C | Sem.C | Spat | Local | Vis.C | Multi-v |
|---|---|---|---|---|---|---|---|---|---|---|
| *Randomly Choose One From the Options* | | | | | | | | | | |
| Random | 50.00 | 25.00 | 50.00 | 50.00 | 25.00 | 25.00 | 50.00 | 50.00 | 25.00 | 50.00 |
| *Randomly Select In-context Exemplars* | | | | | | | | | | |
| 1-shot | 47.41 | 13.33 | 52.42 | 44.67 | 21.54 | 32.37 | 41.96 | 43.44 | 29.65 | 44.36 |
| 2-shot | 47.41 | 2.50 | 54.03 | 52.00 | 22.31 | 32.37 | 38.46 | 43.44 | 29.65 | 55.64 |
| 3-shot | 47.41 | 5.83 | 53.23 | 52.67 | 22.31 | 32.37 | 48.95 | 43.44 | 29.65 | 44.36 |
| 4-shot | 47.41 | 3.33 | 52.42 | 52.00 | 22.31 | 32.37 | 45.45 | 43.44 | 29.65 | 44.36 |
| 5-shot | 47.41 | 17.50 | 54.84 | 50.67 | 22.31 | 30.94 | 45.45 | 43.44 | 29.65 | 44.36 |
| *Select In-context Exemplars Based on CLIP Embedding Similarity* | | | | | | | | | | |
| 1-shot | 47.41 | 8.33 | 56.45 | 51.33 | 21.54 | 28.06 | 39.16 | 43.44 | 24.42 | 45.11 |
| 2-shot | 47.41 | 8.33 | 54.84 | 51.33 | 22.31 | 25.18 | 39.86 | 43.44 | 27.91 | 30.08 |
| 3-shot | 47.41 | 10.83 | 53.23 | 50.67 | 20.77 | 26.62 | 39.16 | 43.44 | 27.33 | 28.57 |
| 4-shot | 47.41 | 10.83 | 52.42 | 51.33 | 23.08 | 29.50 | 39.86 | 43.44 | 27.91 | 33.83 |
| 5-shot | 47.41 | 11.67 | 52.42 | 52.67 | 20.77 | 28.06 | 39.86 | 43.44 | 24.42 | 35.34 |
| *Select In-context Exemplars Based on VIT Embedding Similarity* | | | | | | | | | | |
| 1-shot | 47.41 | 8.33 | 56.45 | 51.33 | 21.54 | 28.06 | 37.06 | 43.44 | 24.42 | 14.29 |
| 2-shot | 47.41 | 8.33 | 54.84 | 50.67 | 22.31 | 25.18 | 38.46 | 43.44 | 27.91 | 30.08 |
| 3-shot | 47.41 | 10.83 | 53.23 | 50.67 | 20.77 | 26.62 | 39.86 | 43.44 | 27.33 | 28.57 |
| 4-shot | 47.41 | 10.00 | 52.42 | 50.67 | 23.08 | 29.50 | 39.86 | 43.44 | 27.91 | 28.57 |
| 5-shot | 47.41 | 10.83 | 52.42 | 52.00 | 20.77 | 28.06 | 41.96 | 43.44 | 24.42 | 34.59 |

Table 13: Few-shot in-context learning results on the Blink dataset using LLaVa-OneVision-7B, evaluated with varying numbers of exemplars and three retrieval methods.

## G FUNCTION DEFINITIONS IN THE VIPACT ALGORITHM

In this section, we provide detailed explanations of the functions used in Algorithm 1, as summarized in Table 14. Each function is essential for coordinating interactions between the orchestrator agent, specialized agents, and vision expert models within the VIPACT framework.

## H PROMPT DESIGN

In this section, we present the complete prompt designs used in our experiments, including the Initial Prompt for the orchestrator agent and the distinct prompt designs for the three specialized agents described in Section 3.

---

**Initial Prompt for Orchestrator Agent**

You are a helpful AI agent and please answer the following question based on the image. You have access to the following tools:
{tools}
Additionally, if you want to use python code, you can use the following functions:

```python
def image_comparison(image_paths: list, focus: str = None):
        '''
        Compares multiple images and generates a detailed
        analysis of their similarities and differences,
        with an optional focus on specific objects, elements,
        or aspects.

        Parameters
        ----------
        image_paths : list
            A list of file paths for the input images to
            be compared.
        focus : str, optional
            The specific objects, elements, or aspects that
            the comparison should focus on.
            If None, a general comparison is generated.

        Example
        --------
            >>> image_comparison(image_paths=["image1.jpg",
            "image2.jpg"], focus="the cars")
        '''
```

---

**Initial Prompt for Orchestrator Agent (Cont'd)**

```python
def image_captioning(image_path: str, focus: str = None):
        '''
        Generates a detailed caption for the provided image,
        with an optional focus on specific objects, elements or
        other perspectives that are directly related to solving
        the problem.

        Parameters
        ----------
        image_path : str
            The file path of the input image.
        focus : str, optional
            The specific objects or elements that the caption
            should focus on. If None, a general caption is
            generated.

        Example
        --------
            >>> image_captioning(image_path="image.jpg")
            >>> image_captioning(image_path="image.jpg",
            focus="a red car and the background buildings")
        '''

    def visual_prompt_describe(image_path: str = "image.jpg"):
        '''
        Analyzes the provided image and describes the specific
        locations and characteristics of various visual prompts

        This function uses a language model to generate a
        detailed description of visual prompts present in the
        image, such as colored circles, bounding boxes, arrows,
        highlights, or textual labels.

        Parameters
        ----------
        image_path : str
            The file path of the input image.

        Example
        --------
            >>> visual_prompt_describe(image_path="image.jpg")
```

**Initial Prompt for Orchestrator Agent (Cont'd)**

```python
def save_depth_image(image_path: str = "image.jpg",
    saved_path: str = "depth.jpg"):
    '''
    Estimates the depth of an input image, saves the
    resulting depth image to a specified path,
    and prints out the saved path in a structured format.

    Note: In the processed depth estimation image, brighter
    areas represent objects closer to the camera,
    while darker areas represent objects farther from the
    camera. For pixel values, higher values (brighter areas)
    indicate closer proximity to the camera, while lower
    values (darker areas) indicate greater distance.

    Parameters
    ----------
    image_path : str, optional
        The file path of the input image.

    saved_path : str, optional
        The file path where the resulting depth image will
        be saved. You should make sure the saved image is
        in the same directory as the input image.

    Example
    --------
        >>>  save_depth_image(image_path = "image.jpg",
        saved_path = "depth.jpg")
    '''

def locate_visual_prompts(image_path: str = "image.jpg"):
    '''
    Analyzes the provided image to identify and accurately
    locate two specific regions labeled 'A' and 'B'.
    This function detects the visual prompts of red circles
    and print out their coordinates.

    Parameters
    ----------
    image_path : str
        The file path of the input image to be processed.

    Example
    -------
        >>> locate_visual_prompts("images/image.jpg")
    '''
```

**Initial Prompt for Orchestrator Agent (Cont'd)**

```python
def compute_clip_similarity(image_path1: str,
    image_path2: str) -> float:
        '''
        Computes the cosine similarity between the CLIP
        embeddings of two images.

        Parameters
        ----------
        image_path1 : str
            The file path of the first input image.
        image_path2 : str
            The file path of the second input image.

        Returns
        -------
        float
            The cosine similarity score between the two image
            embeddings (-1 to 1).

        Example
        -------
            >>> similarity =
            compute_clip_similarity("image1.jpg", "image2.jpg")
        '''

    def segment_image(image_path: str, save_path: str = None)
    -> str:
        '''
        Segments the input image using the SAM model and
        returns the path to the processed image.

        Parameters
        ----------
        image_path : str
            The file path of the input image to be segmented.
        save_path : str, optional
            The file path where the segmented image will be
            saved. If None, a default path is used.

        Returns
        -------
        str
            The file path of the saved segmented image.

        Example
        -------
            >>> segmented_img_path = segment_image("input.jpg",
            "segmented.jpg")
        '''

    # All function implementations are available in the
    execution environment and you can just call the function
    without the need to define it.
        '''
```

**Initial Prompt for Orchestrator Agent (Cont'd)**

MUST strictly use the following format:
Question: [The input question you must answer]
Image: [The path of the image, which you may use in external tools]
Task Requirement: [You should provide a comprehensive analysis of the criteria to choose between each option. Include key factors to focus on in solving this task, such as specific visual elements, data points, trends, patterns, and any contextual information that might influence the decision. You can also try to decompose the problem into several key subproblems, with clues inferred from the following steps.]
Thought: [Your reasoning about the question or the last iteration's observations. You should prioritize to think about which tools to use (and which parameters to input) and if you believe no existing tools will help further, use your own knowledge to reason towards the final answer. If there is no observation from the last iteration's tool calling, you should examine the format of tool calling and recall the tool with proper format]
Action Input: [MUST be some of the functions above within a Python block with nothing else. You should figure out which function to use and what are the input parameters.]
Observation: [The output of the called function.]
... (Repeat Thought/Action/Action Input/Observation as needed, you may need to call the tools multiple times if there are multiple images in the input) Thought: [Your final reasoning based on all information gathered]
Final Answer: [You MUST provide a clear answer from the above options without any ambiguity. If a perfect answer is not available, you MUST select the closest possible option.]
Begin! Let's work on the following question! Please remember NOT to estimate any coordinates in the image within the code.
Question: {question}
Image: {image} Task Requirement: (you should start to generate this to begin the iterations)

**Prompt for Focused Image Captioning Agent**

Please analyze the provided image and generate a comprehensive, detailed caption that focuses specifically on "{focus}". Your caption should:
1. Identify and describe the specified focus objects or elements in the image, including:

- Quantity (the total number of such object)

- Appearance (color, size, shape, texture)

- Position within the image

- Relation to other objects (if applicable)

2. For the focus objects or elements, describe any actions or events taking place, involving any of them.
3. Mention the overall setting or background of the image, especially in relation to the focus.
4. Include relevant details about lighting, shadows, and any visible textures.
5. If there are people or animals in the focus area, describe their appearances, poses, and any visible expressions.
Your goal is to create an extremely detailed and thorough caption that gives a complete understanding of the image's content with an emphasis on the specified focus, as if you're describing it to someone who cannot see it. Don't leave out any visible elements related to the focus, no matter how minor they might seem.
Image: {image}

**Prompt for Focused Image Comparison Agent**

Please analyze the provided images and generate a comprehensive, detailed comparison that focuses specifically on "{focus}". Your comparison should:
1. Identify and describe the specified focus (focus) in all images, including:

- Presence or absence in each image (if applicable)
- Quantity (if applicable)
- Position within each image
- Relation to other objects (if applicable)

2. Compare the overall setting or background of the images, but only as it relates to the focus.
2. Summarize the similarities and differences of the focus elements across all images.
3. Describe any changes in actions, events, or states related to the focus elements (if applicable).
5. Analyze differences in lighting, shadows, and visible textures that affect the focus elements.
Your goal is to create a detailed and thorough comparison that gives a complete understanding of how the specified focus elements differ or remain similar across all the provided images. Concentrate primarily on the focus area and only mention other elements if they directly relate to or impact the focus.
Organize your comparison in a clear, structured manner, addressing the focus area in each image in turn and then providing an overall summary of the similarities and differences.
Image: {image}

**Prompt for Visual Prompt Description Agent**

Please analyze the provided image, emphasizing the specific regions or objects indicated by visual prompts such as colored circles, bounding boxes, arrows, highlights, or textual labels. The most critical aspect of your analysis should be a detailed description of these indicated areas. For each visual prompt:
1. Most importantly, provide an extremely detailed description of the exact region or object being indicated. This is the primary focus of your analysis. Include:

- Precise location within the larger object or scene
- Comprehensive details about its appearance (color, texture, shape, size)
- Any unique features or characteristics
- Its context and relationship to surrounding elements

2. The type of visual prompt used (e.g., circle, box, arrow, highlight, label).
3. The position of the prompt within the entire image (e.g., top left, center, bottom right).
4. Any text or identifiers associated with the prompt (e.g., labels like 'A', 'B', numbers, or short descriptions). Remember, the most crucial part of your response should be the in-depth description of the specific region or object highlighted by each prompt. Provide enough detail that someone could understand exactly what part of the image is being emphasized without seeing the visual prompt itself.
Ensure your description of these indicated regions is as comprehensive as possible, covering every relevant visual aspect. Your goal is to provide a thorough understanding of the highlighted areas, allowing others to easily grasp the significance of each visual prompt in the image.
Image: {image}

**Prompt for Few-shot In-context Learning**

{The general instruction for the task}
Here are some examples:
Images: {example_images}
Question: {example_question}
Answer: {example_answer}
...
Let's try another case!
Images: {images}
Question: {question}
Answer:

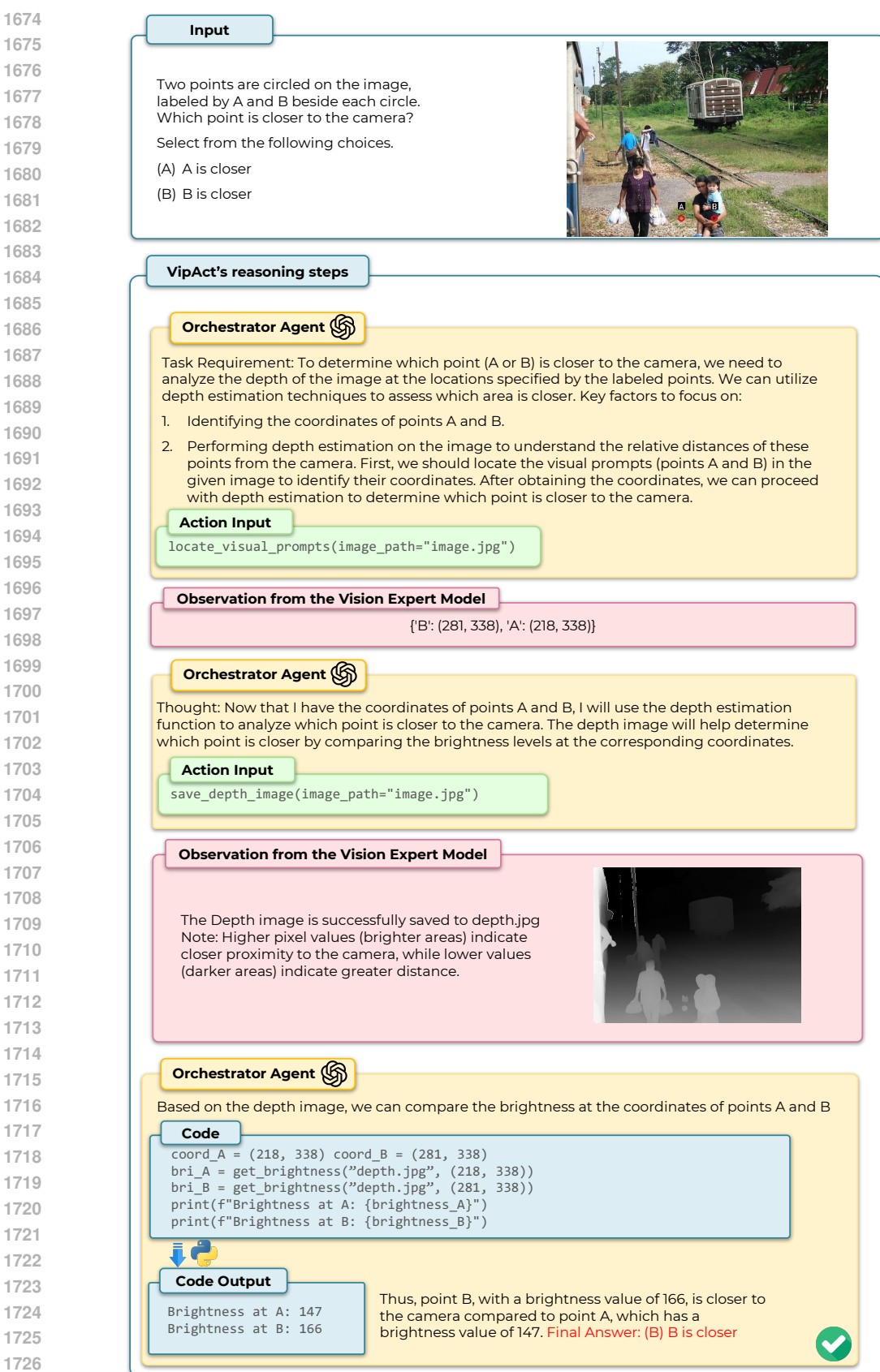

Figure 3: Case study of the complete reasoning process in VIPACT, incorporating a depth estimation model to assist the VLM in achieving the correct answer.

Figure 4: Case study of the complete reasoning process in VIPACT, incorporating a specialized agent to assist the VLM in achieving the correct answer.

| Function | Description |
|---|---|
| FORMATPROMPT($\mathcal{V}, q$) | Combines the visual inputs $\mathcal{V}$ and the query $q$ into a structured prompt suitable for the vision-language model $\mathcal{M}$. This ensures that the orchestrator agent receives a well-organized task description for reasoning. |
| ISTERMINATED($\mathcal{S}$) | Checks whether the termination condition has been met based on the current state $\mathcal{S}$. This involves checking for a termination indicator (e.g., `Final Answer:`) or determining if the maximum number of iterations $K$ has been reached. |
| ISREQUIRED($T_i, \mathcal{S}$) | Determines if a specific tool $T_i$ (either a specialized agent or vision expert model) is necessary given the current state $\mathcal{S}$. This involves checking whether tool-use indicators (e.g., `Action:` or `Action Input:`) have been generated, guiding the orchestrator agent on whether external tools need to be invoked. |
| UTILITY($T_i, \mathcal{S}$) | Implicitly evaluates the utility of tool $T_i$ in the current context defined by state $\mathcal{S}$. This process involves the orchestrator agent select the most beneficial tool for the next action, based on prior evidence and reasoning steps. |
| EXECUTE($T^*, \mathcal{S}$) | Executes the selected tool $T^*$ using the current state $\mathcal{S}$ (arguments extracted from VLM's output at this step) as input. The tool processes the input and returns relevant information, such as image data or analytical results, which are then integrated into the reasoning process. |
| CONTAINSVISUALDATA($\mathcal{O}_t$) | Checks whether the output $\mathcal{O}_t$ from the executed tool includes visual data (e.g., new images or annotations). If visual data is present, it is further processed and incorporated into the reasoning workflow. |
| PROCESSVISUALDATA($\mathcal{O}_t$) | Processes new visual data from the tool's output $\mathcal{O}_t$ and integrates it into the existing set of visual inputs $\mathcal{V}$. This involves updating the prompt with new image paths to ensure that the visual data is available for subsequent analysis and reasoning. |
| INTERPRETOUTPUT($\mathcal{R}_t$) | Interprets the output $\mathcal{R}_t$ generated by the VLM $\mathcal{M}$. This step involves converting the raw output into a structured format through rule-based string manipulation, enabling the orchestrator agent to update the task state and inform the next steps. |
| UPDATEPROMPT($\mathcal{P}_{t-1}, \mathcal{O}_t$) | Updates the current prompt $\mathcal{P}_{t-1}$ with new information derived from the tool output $\mathcal{O}_t$. The updated prompt ensures that the next iteration of the VLM has access to the most recent and relevant context, presented in an organized format for accurate reasoning in the next iteration. |
| UPDATESTATE($\mathcal{S}, \mathcal{O}_t$) | Updates the current state $\mathcal{S}$ by incorporating new observations and data from the tool or VLM output $\mathcal{O}_t$. This continuous state update allows the system to track progress and adjust its strategy dynamically. |
| EXTRACTANSWER($\mathcal{S}$) | Extracts the final answer $a$ from the final output of the VLM. This step uses rule-based string matching to retrieve the final prediction from the agent's workflow. |

Table 14: Function Definitions in Algorithm 1

