# OpenReview forum: "VipAct: Visual-Perception Enhancement via Specialized VLM Agent Collaboration and Tool-use"
_ICLR.cc/2025/Conference — Submitted to ICLR 2025_

### Official Review · Reviewer_Jd2M · 2024-10-26

**Soundness:** 3
**Presentation:** 2
**Contribution:** 3
**Rating:** 6
**Confidence:** 3

**Summary:**

This paper proposes VipAct, a VLM-based agent framework capable of performing challenging visual perception tasks by decomposing the tasks into sub-tasks which specialized agents and expert models then overtake. The VipAct framework is designed as a hierarchical pipeline, with a VLM as the orchestrator agent and specialized VLMs and expert vision models as the tools called by the orchestrator. Unlike existing visual programming methods (e.g., VisProg and MM-ReAct), ViaAct takes the vision task as its input and the input image. This innovation enables VipAct to decompose the high-level task into visually grounded and feasible subtasks that can be submitted to expert models. Ablation studies justify the introduced components, i.e., multiple agents, visual input, and vision experts.

**Strengths:**

1. The authors utilized a VLM-based programmatic framework to perform difficult vision tasks challenging for most existing VLMs.
2. The description of the ViaAct pipeline is clear and easy to understand.
3. The experiments are detailed and comprehensive.

**Weaknesses:**

1. It is still unclear how including the input image in the prompt to the orchestrator agent improves the fine-grained visual perception capability, although the experimental results show that removing this visual input leads to a performance decrease. At the beginning of the paper, the authors stated that "recent studies reveal that state-of-the-art (SOTA) VLMs continue to struggle with fine-grained or low-level visual perception tasks that are trivial for humans" (L43). Based on this premise, the orchestrator agent is also not capable of calling proper tools as which tools with fine-grained perception functionalities are to be called is also based on the fine-grained perception of the input image. The authors are expected to present more vivid examples of how the agent's behavior differs with and without visual input, which would help clarify this point.

2. Although the authors attempt to conduct fair comparisons, one key aspect - the tool sets used by the compared methods - is not fair enough. L400 states that "Another limitation is their inability to support images with visual prompts, preventing them from locating visual prompts and proceeding with subsequent operations". The compared methods were not given the necessary vision expert models to complete the tasks, thereby obtaining inferior results. The reviewer believes that the utilization of the vision experts, such as Visual Prompt Detector, is not part of the paper's innovations and that the compared methods should also be equipped with these vision experts in the experiments. The authors are expected to conduct additional comparisons where baseline methods are equipped with identical tools, or explain why such comparisons might not be feasible or relevant. This would provide a clearer picture of VipAct's unique contributions beyond the use of vision expert models.

**Questions:**

1. What if the authors replace the visual input to the orchestrator agent with a detailed description of it? Can this textual input achieve a satisfactory performance compared with using the original image? The authors may conduct an ablation study comparing performance across three conditions: 1) original image input, 2) detailed textual description input, and 3) both image and textual description as input. This would help isolate the unique contributions of visual versus textual information to the orchestrator agent's performance.

The reviewer would like to change the rating if the concerns are all resolved.

---

> ### Author Response · Authors · 2024-11-26
> **Response to Reviewer Jd2M**
>
> We sincerely appreciate your valuable feedback and the acknowledgment of the **challenges addressed in our work** and the **comprehensiveness of our experiments**. Below, we provide clarifications to address your concerns:
>
> ## W1: Motivation for Including the Input Image in the Orchestrator Agent Prompt
> - Thank you for recognizing the significance of our ablation study. Empirically, we observe that including the input image allows the orchestrator agent to perform **higher-level visual understanding, facilitating better tool selection and specifying more accurate input parameters**. For instance, in a multi-view reasoning task, the orchestrator uses the image input to identify reference objects, enabling it to specify an appropriate input parameter 'focus' for the specialized agent to determine whether the camera is moving left or right.
>
> - Moreover, although SOTA VLMs struggle with fine-grained or low-level perception tasks, they are capable of extracting relatively high-level visual content. This capability enhances **grounded error handling and information aggregation**, especially when specialized agents or tools encounter **execution errors or hallucinate**. Without the image input, the orchestrator cannot perform such error handling, which degrades overall performance.
>
> ## W2: Tool Set Fairness in Comparisons
>
> - We appreciate your comment regarding the fairness of tool comparisons. Existing visual programming methods often **tightly integrate tool usage with their internal logic**, making it challenging to add or modify tools without disrupting their framework. In contrast, VipAct’s modular design enables **plug-and-play** tool integration with minimal effort, requiring only the specification of a standard Python function header, as detailed in Appendix H.
>
> - To address your concern, we conducted **additional experiments** where we integrated our vision expert models into baselines while preserving their original logic. The results shown in the following table indicate minor improvements in tasks such as depth estimation and object localization; however, the overall performance gap between these methods and VipAct remains **substantial**. Empirically, we observed that even with specific tools integrated, **significant errors persisted** due to limitations in the underlying logic of these frameworks. This highlights the importance of VipAct’s cohesive design, which integrates multi-agent collaboration and a structured approach to use vision expert models.
>
> | Condition |   Sim | Count | Depth |   Jig | Fun.C | Sem.C |  Spat | Local | Vis.C | Multi-v | Overall |
> | :-------- | ----: | ----: | ----: | ----: | ----: | ----: | ----: | ----: | ----: | ------: | ------: |
> | MM-ReAcT (with same tools)  |     - | 30.00 |  8.69 |     - |     - |     - | 63.64 |  5.41 |     - |       - |       - |
> | ViperGPT (with same tools) |     - | 29.17 |  3.01 |     - |     - |     - | 48.95 | 23.57 |     - |       - |       - |
> | VisProg (with same tools)  |     - |  3.33 |  5.31 |     - |     - |     - | 31.47 | 18.92 |     - |       - |       - |
> | VipAct    | 81.48 | 70.00 | 90.80 | 68.00 | 61.50 | 60.40 | 86.70 | 63.11 | 91.28 |   62.63 |   73.59 |
>
> ## Q1: Ablation Study on Visual Versus Textual Inputs
>
> - Thank you for this insightful suggestion. We conducted an ablation study comparing three conditions:
>   1. Image input.
>   2. Detailed textual description input:
>      - Generated using GPT-4o with the prompt:
>        “Please generate a detailed description of the following image [IMAGE]”.
>   3. Both image and description as input.
>
> - The results in the following table show that using only detailed textual descriptions **underperforms** compared to using the image input. While adding both image and textual descriptions yields slight improvements in some cases, the gains are **not significant**.  Empirically, we observe that generated descriptions often fail to capture fine-grained details or background elements critical to certain tasks. Furthermore, descriptions can introduce biases, emphasizing objects over contextual elements, leading to suboptimal performance. Overall, this analysis reinforces the importance of direct visual input in enabling robust and grounded reasoning by the orchestrator agent.
>
> | Condition                          |   Sim | Count | Depth |   Jig | Fun.C | Sem.C |  Spat | Local | Vis.C | Multi-v | Overall |
> | :--------------------------------- | ----: | ----: | ----: | ----: | ----: | ----: | ----: | ----: | ----: | ------: | ------: |
> | Image               | 81.48 |    70 |  90.8 |    68 |  61.5 |  60.4 |  86.7 | 63.11 | 91.28 |   62.63 |   73.59 |
> | No image or description input      | 77.78 | 59.71 | 69.35 | 61.33 | 53.85 | 51.08 | 83.22 | 60.66 | 78.49 |   48.12 |   64.36 |
> | Description| 79.32 | 62.72 | 73.45 | 62.37 | 54.02 | 52.46 | 83.22 | 61.34 | 81.35 |   51.42 |   66.17 |
> | Image + description | 81.48 | 70.48 |  90.8 | 67.52 | 62.45 | 61.32 | 84.32 | 62.67 | 91.28 |   63.34 |   73.57 |

---

> ### Comment · Reviewer_Jd2M · 2024-12-01
> **Official Comment by Reviewer Jd2M**
>
> Thank you to the authors for the detailed rebuttal and the expanded experiments.
>
> My concerns have been almost addressed. The experimental results in Q1 are quite interesting.
>
> However, I also agree with the reviewer rhXE that **VipAct did not provide very inspiring innovations** despite its improved pipeline over existing methods like VisProg. But don't worry, I will keep my rating as the authors' experiments and analysis are self-contained, which is also a merit.

---

> > ### Author Response · Authors · 2024-12-03
> > **Response to the reponse of Reviewer Jd2M to our rebuttal**
> >
> > Thank you for your thoughtful review and for acknowledging the expanded experiments and detailed rebuttal. We are pleased to hear that **most of your concerns have been addressed** and that you found the experimental results in Q1 interesting.
> >
> > Regarding your newly raised points, we kindly refer you to our latest reply to Reviewer rhXE, where we have provided additional clarifications that we believe address these concerns. Your feedback is invaluable, and we greatly appreciate your insights.
> >
> > Thank you once again for your constructive comments. It has been a pleasure to discuss our work with you, and we truly value this opportunity for meaningful engagement.

---

### Official Review · Reviewer_rhXE · 2024-10-28

**Soundness:** 3
**Presentation:** 3
**Contribution:** 2
**Rating:** 5
**Confidence:** 4

**Summary:**

The paper introduces VIPACT, a novel framework designed to address fine-grained visual perception tasks, where current Vision-Language Models (VLMs) often underperform. VIPACT utilizes a collaborative multi-agent approach that integrates a central orchestrator agent with specialized agents and vision expert models to achieve more precise and detailed visual understanding.

Contributions:
1. Multi-Agent Collaboration Framework: VIPACT employs an orchestrator agent that manages task analysis, planning, and evidence aggregation, coordinating specialized agents to tackle specific tasks like image captioning and depth estimation. Specialized agents provide focused capabilities (e.g., object detection, depth estimation), enhancing VLM performance in tasks requiring detailed visual analysis.
2. Comprehensive Benchmarking: VIPACT outperforms state-of-the-art baselines on challenging visual perception benchmarks, including Blink and MMVP.
3. Ablation and Error Analysis: Through ablation studies, the paper highlights the importance of each component in VIPACT, while error analysis reveals specific limitations of current VLMs in spatial reasoning and fine-grained visual tasks, underscoring areas for further improvement in multi-modal perception systems.

**Strengths:**

1. The paper is mainly well-written and easy to follow.
2. The paper proposes a new multi-agent framework for fine-grained visual perception tasks.
3. The proposed framework is effective and improves the performance of two datasets over existing baselines.

**Weaknesses:**

1. The framework is tested on only one LLM. Testing on more, e.g., Claude / Gemini, would be more convincing and show the generalization of the framework.
2. The method proposed has limited contribution to the community. There have been multiple papers proposing/applying "LLM with visual tool use" to solve vision tasks. The multi-agent framework has also been verified to be effective on various downstream tasks.
3. The performance of the proposed framework does not seem significant enough. On the Blink dataset, the improvement over CoT is less than 10 %. The improvement in the MMVP dataset is marginal.

**Questions:**

1. Can the authors put an average column in Table 2 to better show the improvements of the method?

---

> ### Author Response · Authors · 2024-11-28
> **Response to Reviewer rhXE**
>
> We sincerely appreciate your constructive feedback and detailed review of our work. Below, we address your concerns and provide clarifications:
>
> ## W1: Tested on only one LLM. Testing on more, e.g., Claude / Gemini, would be more convincing and show the generalization of the framework.
>
> - **Thank you for raising this concern.**
>   - As outlined in **our response to Reviewer cte9**, we have **justified our model selection and conducted additional experiments incorporating Claude-1.5-Snoet and Gemini-1.5-Pro**. These results demonstrate the **robustness and generalizability** of VIPACT across multiple LLMs, addressing your concern comprehensively.
>
> ## W2: Limited contribution to the community, given existing work on “LLM with visual tool use” and multi-agent frameworks.
>
> - **We acknowledge prior work** on LLMs with visual tool use. However, our contribution lies in how these components are cohesively integrated to achieve superior performance and generalization on fine-grained visual perception tasks. Specifically:
>   - **Novelty in Framework Design:**
>     - **Table 1** highlights VipAct's unique features, including direct image input for planning and execution, iterative reasoning, multi-image support, and specialized multi-agent collaboration. These capabilities set VipAct apart from previous visual programming methods like MM-ReAct and ViperGPT, enabling superior performance and generalization in  visual perception tasks.
>   - **Advancements in Generalization:**
>     - While previous visual programming methods rely heavily on predefined tools with limited adaptability (as shown in Table 2), VipAct incorporates a flexible and modular design, enabling it to generalize across diverse tasks.
>   - **Extension of multi-agent interaction to Multi-Modal Domains:**
>     - While multi-agent frameworks have shown promise in text-based tasks, their potential in **multi-modal settings has been underexplored**. Our work bridges this gap by demonstrating the effectiveness of multi-agent collaboration in fine-grained visual perception tasks.
> - We hope this addresses your concerns regarding the novelty and relevance of our contributions.
>
> ## W3: The performance improvement does not seem significant enough (less than 10%).
>
> - **We respectfully disagree with this assessment.**
>   - As stated in **Lines 974-976 of the paper**, we have conducted **significant test** between our proposal and baselines, and the average p-value for comparisons between VipAct and SOTA baselines is 0.006 (<0.01), underscoring the significance of our improvements.
>   - We argue that in real-world applications of fine-grained visual perception, less than 10% performance gains can lead to meaningful advancements, especially on complex datasets like Blink and MMVP.
>   - Furthermore, VipAct demonstrates enhanced generalization and flexibility, offering a robust framework for future extensions in visual perception.
>
> ## Q1: Add an average column in Table 2 to better showcase the method’s improvements.
>
> - Thank you for this suggestion. We **have included an average column in blue in Table 2 in the revised version of the paper**, providing a clearer representation of the overall improvements achieved by our proposal.

---

> ### Comment · Reviewer_rhXE · 2024-11-30
> **Thanks to the reply**
>
> Thanks to the reviewer for the response and the extra experiments. They have addressed some of the concerns. Therefore, I raised my rating.
>
> However, I still think the paper is not good enough for accepted for ICLR (but may be worth being accepted to some other venues) due to 1. limited novelty and contribution. The proposed method basically expands the former visual programming / agentic methods based on new capabilities of current MLLMs like GPT4o and new visual tools like visual prompting. Also, some of the capabilities mentioned in Table 1 are explored in former papers in other tasks. For instance, planning with image input has been applied to web navigation tasks [1]. Iterative reasoning has been proposed by [2] for former VQA tasks. 2. The improvement over the zero-shot baseline on the MMVP dataset is only 2.7%, even with the fine-grained design (which means extra cost on computation and longer response time).
>
> [1] WebVoyager: Building an End-to-End Web Agent with Large Multimodal Models
>
> [2] IdealGPT: Iteratively Decomposing Vision and Language Reasoning via Large Language Models

---

> > ### Author Response · Authors · 2024-12-03
> > **Response to Reviewer rhXE's response to the rebuttal**
> >
> > We appreciate your thoughtful engagement with our rebuttal and your recognition of the additional experiments addressing some of your concerns. We would like to make the following clarifications on the remaining points you raised:
> >
> > - **Limited Novelty and Contribution**:
> >   Thank you for discussing this point. We would like to emphasize that our work specifically targets **visual perception**, which remains a significant limitation of current VLMs, unlike prior works that focus on general reasoning abilities. Additionally, **multi-agent interaction in the vision-language domain** is underexplored. The **substantial gap** in performance and generalization ability **between previous agentic frameworks and ours** highlights the contribution and novelty of our work in advancing fine-grained visual perception. We acknowledge that certain components of our framework may share similarities with high-level ideas from other domains. However, the **significant proportion of novel design elements, along with the substantial performance improvements and robustness in addressing critical limitations of current VLMs**, collectively demonstrate the substantial contribution of our work.
> > - **Only 2.7% Improvement on MMVP**:
> >   We respectfully argue that a 2.7% improvement cannot be considered minimal in the context of fine-grained visual perception tasks. Furthermore, our method consistently **outperforms baselines across a wide range of visual perception tasks in Blink**, which is the primary dataset designed to evaluate visual perception capabilities. This robust performance across diverse tasks underpins the effectiveness of our framework and its potential for advancing the field.
> >
> > Thank you once again for your detailed review and constructive feedback. We truly appreciate the opportunity to engage in this meaningful discussion about our work with you!

---

### Official Review · Reviewer_cte9 · 2024-10-31

**Soundness:** 3
**Presentation:** 3
**Contribution:** 3
**Rating:** 5
**Confidence:** 5

**Summary:**

This paper presents a novel framework aimed at enhancing vision-language models (VLMs) in handling fine-grained visual perception tasks. The VIPACT framework incorporates an orchestrator agent for task management, specialized agents for tasks like image captioning, and vision expert models for detailed visual analysis. This multi-agent approach enables the VLMs to collaborate, utilize specialized tools, and aggregate evidence, leading to improved performance on benchmarks involving intricate visual tasks. Experimental results demonstrate VIPACT's superiority over state-of-the-art models, with ablation studies underscoring the importance of each component. The paper suggests VIPACT as a scalable, flexible system for real-world visual applications​

**Strengths:**

1. The VIPACT framework’s use of an orchestrator agent that coordinates with specialized agents and vision expert models stands out, as it improves VLMs' performance on fine-grained visual perception tasks by enabling collaborative reasoning. This structured, modular approach allows for flexibility and extensibility, making it adaptable for a wide range of tasks.
2. By employing System-2 reasoning, VIPACT goes beyond traditional VLM capabilities, integrating intermediate reasoning steps that are more complex and contextually rich. This approach enhances VLMs' ability to manage detailed visual information, which is crucial for tasks requiring in-depth visual analysis.
3. The paper includes extensive experiments across multiple benchmarks, showing clear performance gains over state-of-the-art methods in diverse visual tasks. These experiments demonstrate the framework's effectiveness and generalization ability, especially in tasks that are inherently challenging for current VLMs.

**Weaknesses:**

1. VIPACT relies heavily on models like GPT-4o for their advanced instruction-following and function-calling abilities. While the framework is adaptable, current results may not generalize effectively to other VLMs lacking these specific capabilities, restricting the framework's accessibility and broader applicability.

2. In MMVP benchmark, how does the proposed model compared with other vision foundation model like llava, internvl, eagle and so on? Eagle also use multi-expert collaboration, can the authors compare between them?

3. Did the authors use other models instead of GPT4-o? like gemini or internvl? If the performance of the proposed model mainly reply on GPT4-o, I will think the contribution is not enough.

**Questions:**

My questions are in the weakness. I will raise my score if my questions are answered.

---

> ### Author Response · Authors · 2024-11-28
> **Response to Reviewer cte9**
>
> We appreciate your thoughtful feedback! We are pleased that you recognize the **novelty of our approach**, the **flexibility and extensibility** of ours, as well as the **extensive experiments** that demonstrate its **effectiveness, generalizability**, and importance in enhancing VLMs' ability to perceive detailed visual information. We want to address your concerns with the following clarifications:
> ## W1/3: Reliance on GPT-4o which may not generalize effectively
> - Thanks for highlighting this! As we acknowledged in **lines 534–537**, our main experiments rely on GPT-4o due to its superior instruction-following and function-calling ability. We **have explored** other VLMs in the original paper, such as LLaVA (details in **Appendix C**), but found that our agent framework can not be effectively applied to this model due to its bad instruction-following ability. As noted in **lines 341–343**, other multi-modal agent frameworks **also rely heavily on GPT-4o/V** [1][2] due to the significant performance gap between it and open-source alternatives. So we argue that building our proposal primarily on GPT-4o is reasonable for advancing cutting-edge research in multi-modal agent systems.
>
> - **Additional experiments:** To further address your issues, we conducted experiments with more VLMs, including **Gemini-1.5-Pro and Claude-3.5-Sonnet**. Results:
>
> ### Claude
>
> | Method        | Sim   | Count | Depth | Jig   | Fun.C | Sem.C | Spat  | Local | Vis.C | Multi-v | Overall |
> | ------------- | ----- | ----- | ----- | ----- | ----- | ----- | ----- | ----- | ----- | ------- | ------- |
> | Zero-shot     | 85.19 | 67.50 | 66.13 | 58.00 | 58.00 | 44.60 | 72.03 | 57.38 | 73.84 | 48.12   | 63.08   |
> | CoT           | 86.72 | 68.33 | 71.77 | 61.33 | 52.31 | 41.73 | 77.62 | 50.00 | 81.98 | 44.36   | 63.62   |
> | LtM           | 87.42 | 67.42 | 68.42 | 59.97 | 58.00 | 45.13 | 73.82 | 57.47 | 74.29 | 47.86   | 63.98   |
> | ToT           | 86.90 | 67.53 | 69.48 | 57.35 | 59.46 | 43.72 | 74.92 | 58.49 | 76.14 | 46.38   | 64.04   |
> | SoM           | 82.65 | 62.78 | 63.81 | 56.79 | 56.73 | 39.58 | 72.00 | 52.47 | 73.74 | 44.63   | 60.52   |
> | VipAct (Ours) | 88.89 | 67.96 | 88.59 | 65.33 | 60.42 | 50.13 | 78.82 | 61.54 | 83.72 | 49.57   | 69.50   |
>
> ### Gemini
>
> | Method        | Sim   | Count | Depth | Jig   | Fun.C | Sem.C | Spat  | Local | Vis.C | Multi-v | Overall |
> | ------------- | ----- | ----- | ----- | ----- | ----- | ----- | ----- | ----- | ----- | ------- | ------- |
> | Zero-shot     | 78.52 | 60.83 | 70.97 | 72.67 | 44.62 | 51.08 | 74.13 | 57.38 | 81.98 | 55.64   | 64.78   |
> | CoT           | 81.48 | 64.17 | 78.23 | 68.67 | 42.31 | 53.96 | 78.32 | 60.66 | 81.98 | 51.88   | 66.17   |
> | LtM           | 84.44 | 62.50 | 75.81 | 73.33 | 46.92 | 50.36 | 73.43 | 63.11 | 84.30 | 54.14   | 66.83   |
> | ToT           | 82.23 | 61.52 | 74.84 | 68.96 | 45.42 | 52.61 | 75.34 | 61.25 | 82.21 | 52.86   | 65.72   |
> | SoM           | 60.74 | 55.00 | 65.32 | 62.00 | 47.69 | 43.88 | 74.13 | 59.02 | 74.42 | 53.38   | 59.56   |
> | VipAct (Ours) | 84.44 | 64.17 | 89.42 | 74.00 | 48.74 | 57.55 | 79.57 | 70.48 | 86.05 | 59.42   | 71.38   |
>
> These results demonstrate that applying our VipAct **consistently outperforms** baselines and highlights the generalization of our approach when used with VLMs possessing strong instruction-following abilities.
>
> ## W2: Results of other models on the MMVP benchmark
>
> - **Additional experiments:** We evaluated Llava-one-vision-7B, InternVL-2-Pro, and Llama-3.2-90b-vision on the MMVP dataset. Results:
> | Method    | LLaVA-OneVision | InternVL-2-Pro | Llama-3.2-90b-vision |
> | --------- | --------------- | -------------- | -------------------- |
> | Random    | 25.00           | 25.00          | 25.00                |
> | Zero-shot | 29.67           | 60.00          | 57.33                |
> | CoT       | 30.33           | 57.33          | 59.33                |
> | LtM       | 30.00           | 58.67          | 57.33                |
> | ToT       | 31.33           | 60.00          | 59.38                |
> | SoM       | 27.00           | 45.33          | 51.33                |
> - These results show that open-source models achieve slightly above random performance. InternVL-2-Pro and Llama-3.2-90b-vision perform better but still lag significantly behind GPT-4o.
>
> - **Regarding Eagle model:** We do not provide the results of this as it was published after the cutoff date (August 28) and is not required for comparison ([Guideline](https://iclr.cc/Conferences/2025/ReviewerGuide)). Moreover, Eagle’s “expert” concept differs from ours: it uses integrated vision encoders, while VipAct incorporates external vision experts without altering the underlying model architecture. We have cited this and **discussed these distinctions** in the updated paper (Lines 110–116).
>
>
> ## Conclusion
> We believe we have addressed the issues and highlighted the updates in blue for clarity in the revised paper.
>
> [1] GPT-4V: A generalist web agent
> [2] WebVoyager

---

> > ### Comment · Reviewer_cte9 · 2024-12-01
> > **Limited novelty and contribution**
> >
> > I agree with Reviewer rhXE that the proposed method basically expands the former visual programming / agentic methods based on new capabilities of current MLLMs like GPT4o and new visual tools like visual prompting. Therefore, I will keep my score.

---

> > > ### Author Response · Authors · 2024-12-03
> > > **Response to Reviewer cte9's response to rebuttal**
> > >
> > > Thank you for your review and for engaging in the discussion of our paper. We believe the **additional experiments** presented in our rebuttal have **directly addressed the concerns raised in your initial review**. For your newly raised points, we kindly invite you to refer to our latest reply to Reviewer rhXE, where we have provided further clarifications that we believe address your concerns. Your feedback is invaluable, and we sincerely appreciate your insights. Thank you once again; it has been a pleasure to discuss our work with you.

---

### Meta-Review · Area_Chair_n5xD · 2024-12-21

**Metareview:**

The AC appreciates the authors' rebuttal and additional experiments, which addressed some concerns and provided interesting insights. However, the paper's contributions are limited in novelty, and AC agrees, as it largely builds on existing visual programming and agentic methods, leveraging capabilities of modern MLLMs like GPT4o and visual tools such as visual prompting. Many proposed features, such as planning with image inputs and iterative reasoning, have been explored in prior works. Additionally, the performance improvement on the MMVP dataset is marginal (2.7% over the zero-shot baseline) despite the fine-grained design, which introduces higher computational costs and longer response times. While the experiments and analysis are self-contained, the incremental contributions and limited improvements do not meet acceptance standards, though the work may be more suitable for other venues after revision.

**Additional Comments On Reviewer Discussion:**

Reviewers engaged in discussion and held their line well with reasons. Kudos!

---

### Decision · Program_Chairs · 2025-01-22

Reject